# Factorio Learning Environment

**Jack Hopkins**[1†]       **Mart Bakler**[1†]       **Akbir Khan**[2]

[1]Independent    [2]Anthropic    [†]Equal contribution

## Abstract

Large Language Models (LLMs) are rapidly saturating existing benchmarks, necessitating new open-ended evaluations. We introduce the Factorio Learning Environment (FLE), based on the game of Factorio, that tests agents in long-term planning, spatial reasoning, program synthesis, and resource optimization. FLE provides exponentially scaling challenges – from basic automation to complex factories processing millions of resource units per second. We provide two settings: (1) *open-play* with the open-ended task of building the largest factory on an procedurally generated map and (2) *lab-play* consisting of 33 bounded tasks across three settings with fixed resources. We demonstrate across both settings that models still lack strong spatial reasoning. In lab-play, we find that LLMs exhibit promising short-horizon skills, yet are unable to operate effectively in constrained environments, reflecting limitations in error analysis. In open-play, while LLMs discover automation strategies that improve growth (e.g electric-powered drilling), they fail to achieve complex automation (e.g electronic-circuit manufacturing). We have released FLE as an open-source platform[1].

## 1  Introduction

Large Language Models (LLMs) have demonstrated remarkable capabilities at solving complex question-answer (QA) problems, saturating benchmarks in factual recollection (Hendrycks et al., 2021), reasoning (Cobbe et al., 2021) and code prediction (Chen et al., 2021).

The strong performance across these diverse tasks suggests that LLMs have developed sophisticated reasoning capabilities, leading researchers to explore whether models can act as autonomous agents (Yang et al., 2023). This has motivated a number of new agentic benchmarks focusing on long-term planning (Liu et al., 2023; Ruan et al., 2023), learning in complex environments (Paglieri et al., 2024; Jimenez et al., 2023) and reliably learning from mistakes (Xing et al., 2024; Yamada et al., 2023; Kambhampati et al., 2024). However, similar to QA settings, these agentic benchmarks are likely to face saturation due to their natural completion states; which impose an upper bound on performance and limit our ability to differentiate superhuman models.

We introduce the **Factorio Learning Environment** (FLE): a novel evaluation framework built upon the game of Factorio that uniquely addresses this limitation by enabling unbounded agent evaluation with no natural completion state. In this environment, agents must navigate rapidly scaling challenges from basic resource gathering to complex automation while managing an exponentially scaling technology tree - creating natural curricula for evaluating increasingly capable agents.

Agents are tasked with producing factories, whose performance is measured through production throughput, which ranges from early-game rates of $\sim$30 resources/minute to advanced systems processing millions of resources/second. This enables us to meaningfully differentiate agents by

---

[1]https://github.com/JackHopkins/factorio-learning-environment

39th Conference on Neural Information Processing Systems (NeurIPS 2025) Track on Datasets and Benchmarks.

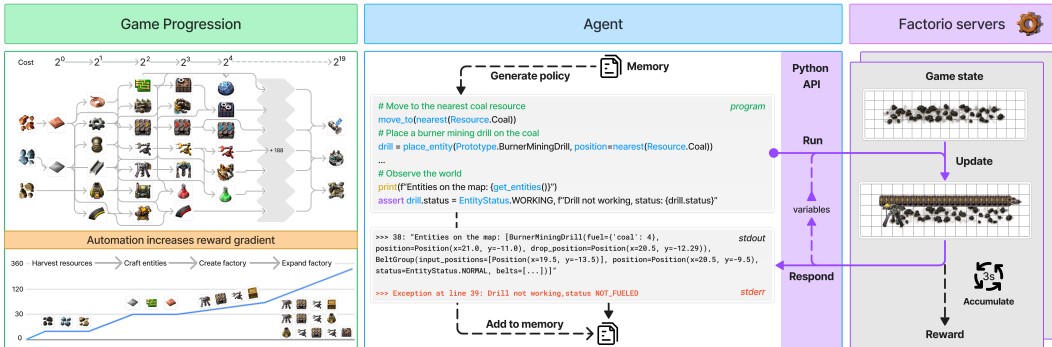

Figure 1: **Illustration of the Factorio Learning Environment (FLE)**. FLE is based on the popular construction and management simulation game *Factorio*. Left: The open-ended goal of the game is to create the largest factory possible. The game enables agents to invest in technological advances to unlock more efficient items and produce more resources per second. Middle: Agents interact with the game by using an interactive Python Interpreter, where they take actions and print their observations in a Read-Eval-Print loop. By using the Python namespace, agents may store variables and define functions for later use. We provide a Python API to Factorio which allows direct interaction with the environment. Right: The agent may issue commands to the game server in order to interact with the environment (with associated time penalties), and receive a response as feedback. If the agents chooses, it may view its own production statistics.

measuring the order of magnitude of resources that they can produce, avoiding saturation by agents even as models become dramatically more capable.

Existing resource management environments such as Minecraft (Guss et al., 2019) or Nethack (Küttler et al., 2020) do not demand the precise industrial optimization present in Factorio. For resource processing chains, producing basic electronic circuits (an early-game staple) requires coordinating 10+ machines processing approximately 15 items per minute. For example, a single rocket component requires orchestrating 60+ interlinked machines manufacturing 1000+ items per minute. The precision required, where a single misaligned machine can cause a factory-wide gridlock, creates a natural curriculum, testing both basic automation and advanced system optimization.

Agents interact with the FLE by synthesizing Python programs to alter and observe the game state, using the tools included in the environment in a Read–Eval–Print Loop (REPL). This feedback loop mirrors the day-to-day workflow of human programmers, who write provisional code to probe how systems behave, interpret the results, then refine their mental model of the system.

We evaluate six frontier LLM models in this environment in an agentic setting. In our qualitative analysis, we study the agents capabilities for spatial reasoning, long-term planning, and error correction. Our results show that even the most advanced models struggle to coordinate more than 10 machines when automatically producing items with over three ingredients.

We summarise our contribution as follows:

- The introduction of the Factorio Learning Environment, an agentic evaluation of long-term planning and resource management.

- Evaluation of frontier models in the unbounded FLE open-play setting in a full Factorio game map. We find more capable agents set and follow longer horizon objectives and achieve quantitatively different production gradients on a log-reward, log-step graph.

- Evaluations of frontier models in FLE lab-play setting, including a set of 24 bounded tasks requiring agents to build factories with increasing complexity and scale. Claude 3.5-Sonnet (the best model) only completes 7/24 tasks and shows limitations in spatial planning in more complex objectives; demonstrating large head-room for performance.

- A qualitative analysis of the results across capabilities such as error-correction and long-term planning. We identify a gap in models' ability to perform intelligent error correction, conduct spatial analysis and set long time-horizon objectives.

```
1  # 1. Get iron patch and place mining drill
2  drill = place_entity(entity=Prototype.MiningDrill, position=nearest(Resource.IronOre)),
       direction=Direction.NORTH
3  )
4  # 2. Add output storage
5  chest = place_entity_next_to(entity=Prototype.IronChest, reference_position=drill.drop_position,
       direction=Direction.SOUTH
6  )
7  # 3. Verify automation chain and observe entities
8  assert drill.status == EntityStatus.WORKING
9  print(get_entities())
```

Figure 2: **Example of an FLE program** used to create a simple automated iron-ore miner. In step 1 the agent uses a query to find the nearest resources and place a mine. In step 3 the agent uses an assert statement to verify that its action was successful.

## 2 Factorio Learning Environment

Our main contribution is the release of an open-source framework, which includes i) a high-level Python API and client to Factorio, ii) a persistent coding environment for LLM agents to interact with the game through iterative program synthesis, and iii) a Python object model of game entities. The environment is procedurally generated, deterministic at runtime (set by a random seed) and is $4 \times 10^{12}$ square tiles in size. We provide a laboratory environment with accessible resources for benchmarking agents in a controlled setting.

### 2.1 Environment Dynamics

Factorio is a resource management and automation game in which players spawn on a world containing raw resources such as water, iron ore, and coal, and must orchestrate increasingly complex production and logistic chains to ultimately produce a rocket and (optionally) escape. The game contains 212 entity types, with a technology tree that unlocks more efficient buildings, resource production chains and multiplicative throughput bonuses. Research enforces a steep resource progression, with late-game technologies such as the `rocket-silo` demanding 300 times more resources than early `automation` research[2]. Player strategy and factory architecture evolves dramatically as technology progresses from early-game manual crafting and basic automation to late-game, massively parallelized, distributed and high-throughput designs.

### 2.2 Environment Interface

Agents interact with FLE through a **REPL** (Read-Eval-Print-Loop) pattern, observing the current game state via previous program output streams, then generating and executing Python code to implement their intended actions, and finally returning useful feedback for the next iteration.

Agents are provided with the Python standard library, and an API comprising methods designed to balance expressiveness with tractability (see Appendix E.1). These initially comprise 10 observation methods and 13 action methods. Observation methods (e.g `nearest`, `get_entities`) retrieve information about the environment, and action methods (e.g `move_to`, `craft_entity`) modify the environment. Each method returns a typed object (e.g an `Inventory`) which can be stored as a variable in the Python namespace and referenced later in the episode. The namespace acts as an episodic symbolic memory system, and saved objects represent part of the environment at the moment of query. This design enables agents to maintain complex state representations and build hierarchical abstractions as the factories scale.

Agents observe **stdout** and **stderr** - the output streams of their program. Thus, agents may intentionally print relevant objects and computations to the output stream to construct observations. Mistakes in the code or invalid operations raise typed exceptions with detailed context that is written to **stderr**. This enables agents to *reactively* debug their programs after execution, and *proactively* use runtime

---

[2]This progression approximately follows an unbounded geometric relationship between resource cost $C$ and research tier $N - C[N] = 1000 \times 2^{(N-1)}$

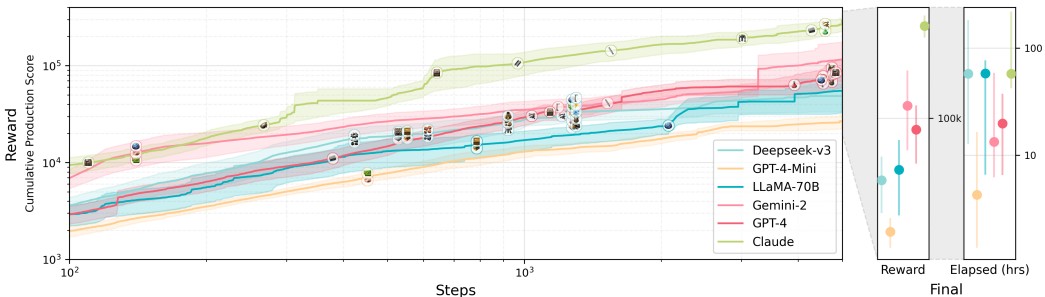

Figure 3: **Models are differentiated by score in Open-Play.** Agents are given the instruction to *build the biggest possible factory*. Left: We find that by evaluating PS against steps (server calls) we can clearly differentiate stronger models from weaker ones in a log/log projection. We overlay milestones, showing the first time the median agent was able to create a new type of entity. Right: We plot the final reward and elapsed game time after 5k steps. We find that while weaker models show promise early-game, they struggle to progress when automation and logistics are required. We report median and standard error over the independent runs.

assertions during execution to self-verify their actions. Programs that take too long to execute are terminated, to prevent runaway control flows (e.g while True).

An environment "step" is a single submission to the Factorio server, which returns the stdout, stderr, rewards and in-game wall-clock time (see Figure 1). Agents are able to enhance their internal representation of the game state in 2 ways: (i) they can define utility functions for reuse throughout an episode, to encapsulate successful logic; and (ii) they can define classes in the namespace to better organize the data retrieved from the game.

The Python API provides complete access to positional data, entity relationships, and geometric constraints, enabling precise spatial reasoning through coordinate systems. For complex factory planning, coordinate-based reasoning offers distinct advantages: exact positioning, systematic layout algorithms, and explicit geometric relationships. For example, diagnosing why a mining drill at (10, 5) fails to output ore requires querying the belt at drop position (10, 4), recognizing its directional misalignment, and inferring the geometric constraint violation - the same reasoning engineers apply to circuit layouts and network topologies. Our troubleshooting experiments (Appendix B.2) support this: visual input provided no performance improvement, confirming that reasoning capability, not information format, determines success.

## 2.3 Reward Structure

We use Factorio's built-in production tracking system, which enables us to define two complementary reward signals:

**Production Score (PS):** A continuous measure of economic activity based on the value of all items produced. This metric increases as agents refine raw ores into manufactured goods and create automatic factories. As factory throughput scales exponentially, PS can vary by multiple orders of magnitude (a rocket launch requires $\approx 10^7$ raw resources). PS provides a naturally unbounded measure of performance, which is sensitive to increasing automation. The game's price calculation system assigns higher value to items with more complex production chains, creating a reward structure that encourages sophisticated factory designs. For the full pricing system, see Appendix A.

**Milestones**: A discrete set of achievements for producing novel item types (e.g. building an `inserter` for the first time, assembling `electronic-circuits`, etc.) and researching technologies. This captures both the diversity of an agent's exploration across Factorio's tech tree, and what level of item complexity they were able to achieve. As Factorio supports unlimited technology research with multiplicative bonuses, milestones can be used to measure performance at all levels of capability.

## 2.4 Implementation Details

The FLE comprises a Python client and Lua server communicating synchronously via RCON over TCP[3]. The client provides the stateful environment interface and APIs, while the server manages game state execution in the official Factorio multiplayer server. The server can be run in headless mode for efficient parallelization. The object model represents most early to late-game entities (detailed in Appendix E.1). FLE is compatible with v1.110 of Factorio, and requires a single purchased game license, as each server must be "activated" by any official client at startup. Fixing the game version also offers a deterministic stable environment, crucial for a reproducible benchmark. FLE is also easily extensible by the community. Designing new tools requires implementing a client-side controller (Python) and a server-side action (Lua) which will automatically load and update the API schema for subsequent agent runs.

## 3 Experiments

To evaluate agent capabilities in FLE, we introduce two settings and a simple agent scaffolding. Open-play comprises of an open-ended sandbox with an unbounded objective and lab-play creates a set of tasks with predefined objectives. All experiments are run in a single-agent setting.

### 3.1 Open Play

In *open-play*, we evaluate each agent in a purely open-ended, unbounded setting. The agents spawn into a procedurally generated world with unlimited space and resources, allowing the agents to decide how best to advance in the game. To progress long-term, agents must show proficient long-term goal-setting, entity and resource planning and spatial reasoning capabilities when creating automated structures. Agents must be capable of using the API, querying the environment for unknown information and reasoning over observations to plan successfully.

We use two metrics to evaluate progress in the game: *Production Score* (PS) and *Milestones*. While the PS acts as the reward and is affected by exploitation, milestones give an overview of how much of the game and technology tree the agent has explored. Each agent plays until the maximum trajectory length of 5000 is reached. This allows us to assess competency within a bounded timeframe as 5000 steps correspond to multiple in-game hours and balances computational cost with sufficient temporal horizon for meaningful agent behavior to emerge. After every agent step, the production throughput is tracked and reward computed. We execute 8 independent runs for each agent, and report the median.

### 3.2 Lab-Play

In *lab-play*, we test agents in planning and spatial reasoning over an environment with abundant resource availability, a full inventory and a completed technology tree. We evaluate two task types with clear completion states:

**Planning Tasks** require the agent to create a factory with a specific production throughput in the constrained lab environment. These tasks are designed to evaluate planning and foresight in designing systems that operate sustainably beyond the duration of an episode.

The objective is to build fully automatic production lines of 24 increasingly-complex distinct target entities, starting from a single resource mine requiring at most 2 machines (making `iron-ore`) to a late game entity requiring the coordination of close to 100 machines of various types (making `utility-science-pack`). The task difficulty increases geometrically with the scaling resource requirements of harder target entities. This makes completing the last task approximately $30\times$ harder than the first. Additional information on task throughputs and complexity can be found in Appendix H.

Each task runs a trajectory of *128* interactions. After every agent step, the throughput of the created structure is evaluated throughout a 60 second holdout period in-game, and the task is deemed completed if the throughput of the structure is above the target throughput at any step $i$. The target throughput is *16* for solid items (for instance electronic circuit, military science pack, plastic bar) and *250* for fluids (for instance petroleum gas, lubricant, heavy oil) during the holdout period. We report

---

[3]Roughly 80k LoC in total

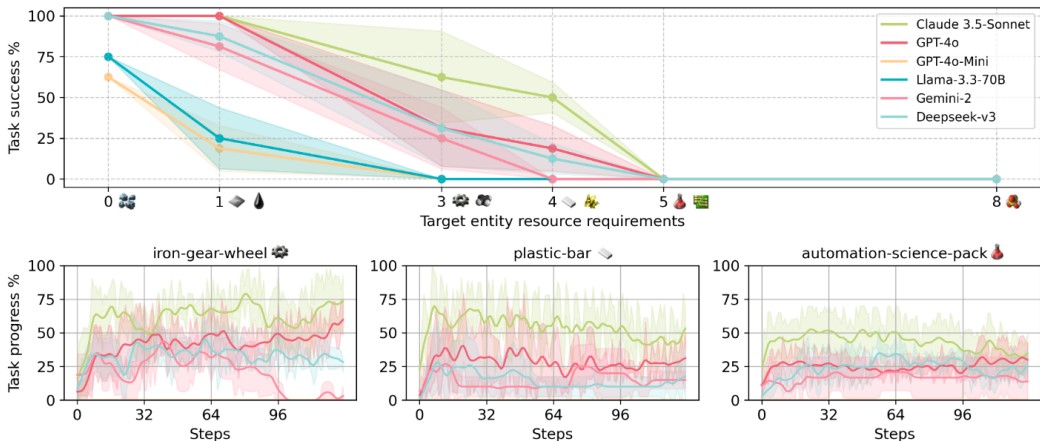

Figure 4: **Agents struggle to build and maintain complex and integrated factories in Lab-Play**. Top: We measure the mean and standard deviation of task success rates across the first 8 complexity levels. We observe a clear decrease in average task success rates as the crafting complexity of the target entity increases. Bottom: We show the mean and std of task progress (percentage of target ingredients and its sub-ingredients agents factories produce at each time-step) in three tasks of increasing difficulty across 8 runs per task. In harder tasks, agents show trends of initial rapid progress followed by stagnation or decrease. This is due to agents being unable to scale up initial production or add new sections to factories required to successfully reach the target production levels and often breaking existing structures during the process. The lack of consistent progress is also observed through the large variance in task progress across runs.

the mean and std of success rates across 8 runs per task. We also report the human baseline, which one of the authors managed to achieve with a reasonable time-frame (32 steps).

**Troubleshooting Tasks** test agents' ability to reason about spatial relationships and factory configurations through two modalities - text and vision. For LLM agents using the API, we evaluate their troubleshooting abilities - inspect human-built factories (15-40 entities) and identify spatial errors that render them inoperable: removal, duplication, rotation, and offset errors. Each task runs for 16 interactions, during which agents can inspect and analyse the factory before submitting discovered errors (e.g., (0,5)=ROTATION).

**Visual Spatial Reasoning** For VLM agents, we also evaluate four visual reasoning tasks on rendered game images, where each task uses multiple-choice questions with distractors designed to be plausible but incorrect. We report classification accuracy for API-based agents (F1 score), accuracy scores for VLM agents, and human baselines across all tasks.

### 3.3 Agent Scaffolding

We consider a simple step-by-step prompting approach as a baseline implementation for agents to interact with the environment. The input prompt of the agent consists of the API schema $A$, a guide $G$ describing how to use the API tools with code examples and the memory $M$ of the agent consisting of past policies with environment observations. A detailed description for the guide, API schema and an example memory state is exhibited in Appendix K. Given the inputs, the agent is tasked to identify the most useful next step and generate the Python policy $P$ that carries out actions in the environment to achieve the step. The policy is executed in the environment and added to the memory $M$ with the environment observations (**stdout**) and error messages (**stderr**). The updated memory $M$ is used as input to generate the next policy and enables the agent to gather information from the environment and use observations to guide future step generation.

**Long context memory**. While Lab-Play tasks are short (128 steps) and do not require memory management to succeed, open-play trajectories can span hundreds of hours and require memory access over thousands of steps. To limit the memory token count and avoid long-context performance degradation, past observations and policies older than 16 steps are summarised by the agent into a

report of 1024 tokens. The report captures errors, solutions and learned best practices from agent interactions with the environment. This allows execution of arbitrarily long traces in the environment without intractable input token requirements. The report also includes the function signatures (name, input/output types and description) of any agent-defined functions, for future composability and code reuse. This allows the agent to develop abstractions to encapsulate and reuse common operations and routines (e.g., factory area placement, raw resource gathering, environment exploration). The agent can also store state in the Python namespace (e.g., entities, coordinates, inventories) and cache intermediate results. These measures are taken to mitigate pressure on the context window and avoid U-shaped performance degradation over long-contextsLiu et al. (2024b). The agent is informed of this summarisation procedure and its own capability to handle memory.

**Language Models** - We evaluate frontier closed source models including Claude 3.5-Sonnet Anthropic (2024), GPT-4o and GPT-4o-Mini OpenAI et al. (2024), Deepseek-v3 DeepSeek-AI et al. (2025) and Gemini-2-Flash Team et al. (2024). We also evaluate Llama-3.3-70B-Instruct MetaAI (2024). Each model is sampled at temperature $0.5$. Model timestamps are in Appendix F.

## 4  Results

We analyse agent performance during *open-play* and *lab-play*, and observe common patterns amongst trajectories from both settings. We report experimental costs in Table 6.

**Open-play**: In *open-play*, Claude 3.5-Sonnet outperforms other models in both median PS (293 206) and milestone count (28), surpassing the early-game resource extraction phase and partially investing in technology research (see Figure 3). In comparison, GPT-4o and Gemini-2 Flash made initial progress but did not develop production lines of > 5 entities, and struggled with both creating complex structures and scaling up existing production. Llama 3.3, GPT-4o-Mini and Deepseek-v3 created only trivial structures and mostly preferred manual crafting. In terms of game progression, all factories created were at the early-game range ($3 \times 10^4 \leq PS \leq 2 \times 10^5$), compared to the requirement for a end-game rocket launch of $PS \approx 1.2 \times 10^8$, or the Factorio human record of $PS \approx 3 \times 10^{12}$ (ExEvolution, 2024).

**Lab Play**: For *lab-play* planning tasks, Claude 3.5-Sonnet performed the best, solving 7/24 tasks and managing to create automatic structures typically seen in Factorio's early game; specifically, compact drilling lines coordinating 10+ machines across up to four factory sections (see Table 1). GPT-4o (5/24 tasks solved), Deepseek-v3 (5/24) and Gemini-2-Flash (4/24) managed to consistently create simpler factories but struggled with higher complexity tasks (See Figure 4). Llama-3.3-70B and GPT-4o-Mini (both 2/24 tasks solved) were only able to create single machine factories.

In general, agents were only able to solve low-complexity tasks (see Figure 4). The hardest solved task (`plastic bar`) has a complexity measurement of 9.4, whereas the hardest overall *lab-play* throughput task (`utility-science-pack`) has complexity measurement of 374.8, i.e ∼39 times more complex than the hardest solved task (for full complexity measures, see Appendix H). In comparison, a human baseline was able to solve 20/24 tasks within 128 steps (See Appendix H.7), demonstrating a significant capability gap. For *lab-play* LLM troubleshooting tasks, Claude 3.5-Sonnet performed best (0.26 F1), followed by GPT-4o (0.10 F1) and Deepseek-v3 (0.09 F1), with no improvement when using the vision modality. For VLM spatial reasoning tasks, models achieve between 50-80% of the human baseline, demonstrating a significant capability gap (see Appendix B.2).

### 4.1  Analysis

**Insight 1: Agents who carry out long-term planning do better in open-play**. In *open-play*, agents are given an open-ended directive to create the biggest factory possible, and need to set instrumental objectives themselves to make long-term progress. We observe a clear trend that agents who showed higher long-horizon planning capabilities combined with ambitious objective-setting achieve a higher PS in *open-play* (See analysis at Appendix B.1). This is highlighted by a discrepancy between *lab-play* and *open-play* results, where Deepseek-v3 succeeds in lab-play with early-automation (see Figure 4) but rarely attempts to create factories in open-play without a set objective, resulting in weak entity crafting statistics (See Figure 5) and poor production scores. This shows that the ability to set useful long-horizon objectives is independent from the ability to successfully use and interact with novel APIs in simulated environments.

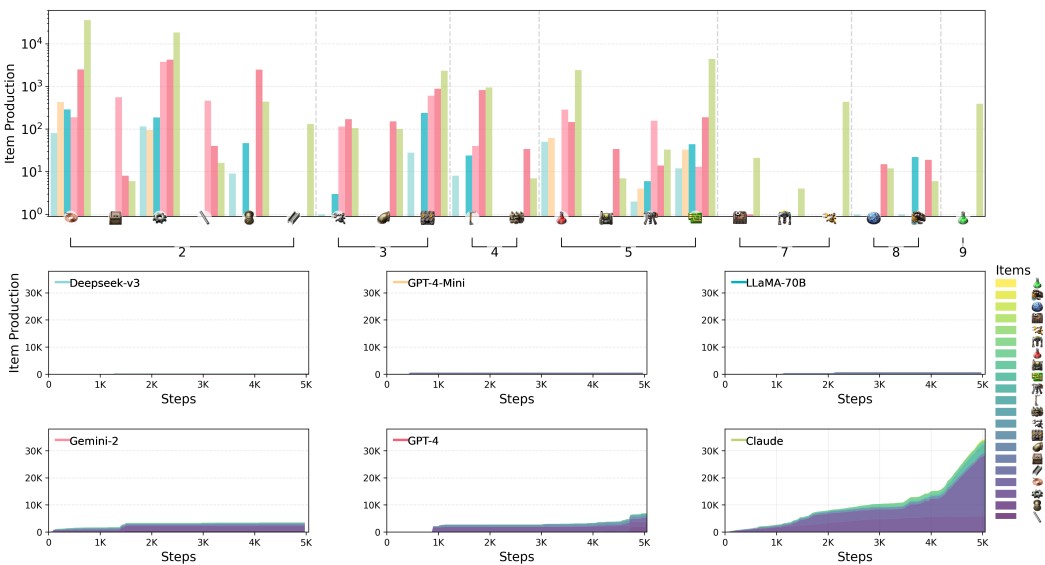

Figure 5: **Open-ended challenges highlight differences in objective setting and general capability.** We illustrate the rates at which various models produce items with multiple antecedent ingredients in the *open-play* setting across number of distinct antecedent ingredients required to craft an entity (top) and accross steps (bottom). Claude 3.5-Sonnet immediately begins complex crafting and invests in research, ultimately unlocking `electric-mining-drills` at step 3000, the deployment of which boosts production of `iron-plate` thereafter. Weaker models like GPT-4o-Mini produce insignificant quantities of multi-ingredient items.

Table 1: **Model performance and coding characteristics in bounded Lab-Play**: This table combines task performance with code analysis for different models. Planning task success rate measures completion percentage across 24 structured tasks. Spatial task accuracy measures the best performing VLM*/LLM agent. Code characteristics include average lines per program (L), percentage of lines that were print statements (P%), percentage of lines that were assertions (A%), and percentage of programs that failed (F%). For failed programs, we track the error types: assertion fails (AF%), code errors (C%), and environment errors (En%).

| Model | Planning task success (%) | Spatial task accuracy (F1) | Code Characteristics | | | | Error Types (%) | | |
|---|---|---|---|---|---|---|---|---|---|
| | | | L | P% | A% | F% | AF% | C% | En% |
| Claude 3.5* | $21.9 \pm 1.3$ | 0.24 | 65 | 43.3 | 2.0 | 50.6 | 0 | 3 | 97 |
| GPT-4o* | $16.6 \pm 1.4$ | 0.10 | 81 | 10.3 | 12.8 | 10.2 | 2 | 12 | 86 |
| DeepSeek-v3 | $15.1 \pm 1.7$ | 0.09 | 37 | 25.4 | 13.9 | 25.3 | 0 | 2 | 98 |
| Gemini-2 | $13.0 \pm 1.3$ | 0.08 | 133 | 16.2 | 0.0 | 16.6 | 1 | 46 | 53 |
| Llama-3.3-70B | $5.2 \pm 1.0$ | 0 | 38 | 23.9 | 12.9 | 23.7 | 0 | 24 | 76 |
| GPT-4o-Mini* | $4.2 \pm 0.6$ | 0.04 | 77 | 36.0 | 0.0 | 31.6 | 15 | 6 | 79 |

**Insight 2: Agents lack spatial reasoning to iteratively increase factory complexity.** A key characteristic for success in Factorio involves iteratively combining multiple factory sections to create complex production lines. In the *lab-play* throughput tasks, we show in Figure 4 that success rate is inversely proportional to target-entity crafting recipe complexity (and thus proportional to the complexity of the required factory). Frequent source of failures when constructing complex factories include trying to place entities too close or on-top of each other, not leaving room for connections and incorrect placement of inserters. This is also illustrated by the results for *lab-play* spatial reasoning tasks, in which frontier models both overlook clear spatial errors in their factory, and hallucinate structural issues where none exist. Interestingly, we note that there is no improvement in a VLM setting, where the agent is given direct images of the factory (See Appendix B.2). We conclude that weak spatial reasoning presents a major bottleneck for agents in FLE.

**Insight 3: Agents display limited error-correction abilities and fall into degenerate debug loops.** A critical component for successful runs was an agents' ability to interact to previous error logs

Table 2: Unbounded Game Environments

| Environment | Crafting Depth | Action | Observation | Agent Type | Open Ended |
|---|---|---|---|---|---|
| **Factorio (FLE)** | **19** | Code (Py) | Symbolic/Visual | LLM | ✓ |
| Voyager | 6-7 | Code (JS) | Symbolic | LLM | ✓ |
| MineDojo | 6-7 | Discrete (300+) | Multimodal | RL/LLM | ✗ |
| NetHack (NLE) | 2-3 | Discrete (93) | Symbolic | RL | ✗ |
| Crafter | 2-3 | Discrete (17) | Visual | RL | ✗ |

and carry out error correction. In *lab-play*, in successful task completions, 56% of steps resulted in program execution errors (from which agents recovered), and in *open-play* this ranges from 29.7% to 76.4%. Claude 3.5-Sonnet, GPT-4o and Deepseek were capable of simpler error correction but lacked the ability to debug complex environments containing subtle defects, as side-effects introduced during debugging destabilise the environment. In *lab-play*, this limitation is illustrated by the frequent decrease of task performance across steps in Figure 4 where the agents broke existing working structures due to incorrectly identifying the root-cause of problems. In *open-play* this results in the flat-line behaviour seen in Figure 3 with no PS progression. This was often due to agents falling into a loop of greedily repeating the same fix rather than exploring additional potential sources of the problem. For instance, in one run GPT-4o used the same API method incorrectly for 78 contiguous steps (from Step 120), receiving identical error message each time [4].

**Insight 4: Agents exhibit different coding styles while interacting with the environment.** We evaluate trajectories with automatic checkers to evaluate how successful models are at using the FLE API. We find that models exhibit different coding styles, with GPT-4o using more assert checks (defensive programming) in within their code than Claude 3.5-Sonnet, which favors a REPL approach with high print usage. Notably, Gemini-2 produces the longest programs but makes the most code errors, while GPT-4o-Mini has the highest rate of assertion failures. These suggest models use very different approaches to explore and engage with the environment in FLE. Using prints suggests being uncertain of state, and exploring new areas, whereas assert statements are likely used to clarify existing knowledge (see Table 1).

## 5 Related Work

Games have served as fundamental benchmarks for AI research, providing standardized environments with clear metrics and natural difficulty gradients (Campbell et al., 2002; Silver et al., 2016; Berner et al., 2019). Recent advances have increasingly focused on using LLMs within game-like environments, exploring their interactive and agentic capabilities.

**Game Environments.** MineRL (Guss et al., 2019), MineDojo (Fan et al., 2022) and Voyager (Wang et al., 2023) provide agent interfaces to Minecraft, an unbounded and open-ended crafting game. ALFWorld (Shridhar et al., 2020) combines language understanding with embodied closed-ended tasks in bounded household environments. NetHack Learning Environment(NLE) (Küttler et al., 2020) offers an unbounded, highly complex, long-term planning and resource management (5,976 distinct entities) environment based on NetHack, which was adapted for LLM usage by (Jeurissen et al., 2024). Finally, BALROG (Paglieri et al., 2024) provides a framework to benchmark LLMs on six RL-environments spanning visual puzzles to real-time strategy, focusing on evaluating navigation, exploration, resource management and long-term planning.

**Resource Automation.** While several environments test manual resource management by an agent: Crafter (Hafner, 2021), NetHack, MineDojo, and Voyager; only FLE challenges agents to maintain automated systems of resource production as a primary emphasis. Moreover, the exponential scaling requirements of resource automation provides an evaluation signal that avoids saturation by models even as their capabilities scale by orders of magnitude (See Appendix J for example derivation of complexity comparison between FLE and modded Minecraft).

---

[4] On two occasions, GPT-4o-Mini simply gave up and repeatedly asked to be reset - see Appendix G

**Factorio for AI Research.** Prior work has explored Factorio as a platform for AI research (Kant, 2025), and for closed-domain settings (Reid et al., 2021), focusing on integer programming models and meta-heuristics. We build on this foundation to offer a standardized text-based interface for agents to solve open-ended challenges in long-term planning, spatial reasoning, and factory optimization.

# 6    Limitations, Future Work & Conclusion

A major concern for any environment benchmark is reward hacking (Clark & Amodei, 2016; Skalse et al., 2022). In our setting, this could involve two main attack surfaces: Python API (as seen within Denison et al. (2024)) or within the Factorio game-engine itself via malicious Lua code submission. During our evaluations, the agent was able to occasionally trigger resetting the Factorio gamestate but we observed no direct examples of reward hacking.

While our Python API sidesteps visual processing, a promising direction for future work is developing a GUI-based interaction track. Such a track would enable comparison of agent performance across different interaction modalities and could reveal whether certain spatial tasks benefit from visual-motor interfaces.

Whilst the authors were, in shorter game time, able to outperform frontier agents, it is unclear if achieving end-game goals (e.g. escape the world) is achievable to humans using only an API in a reasonable time-frame. We did however prove that each step in the chain to launch a rocket was achievable, and that all tasks in *lab-play* can be completed. Moreover our *lab-play* error detection experiments showed limited or no improvement when including vision as an additional modality. Additional improvements to observation space could include more sophisticated error correction and debugging support, inspired by code refinement techniques from Liu et al. (2024a).

While our work currently focuses on single-agent interactions, FLE also supports multi-agent scenarios. A promising extension includes cooperative and competitive multi-agent scenarios, inspired by frameworks like Camel Li et al. (2023), Project Sid AL et al. (2024), and Generative Agents Park et al. (2023). This would enable exploration of emergent cooperation, competition (for finite resources), and multi-objective optimization.

Finally, FLE was intentionally designed to support sophisticated agent scaffolding. We have made it simple for researchers to plug in their own more advanced memory systems, multi-agent protocols, tool-use, planning abstractions, and have exposed a Gym-style interface to the environment. Nevertheless, all of our insights and findings are centered around reasoning behaviour and not affected by agentic scaffoldings. We fully encourage alternative agent implementations and leave that for future work.

**Conclusion**: In this work, we introduce the Factorio Learning Environment (FLE), a novel framework for evaluating the capabilities of agents in an unbounded, open-ended environment. The unbounded nature of FLE provides a benchmark that will resist saturation as progress in LLMs continues to advance. FLE's exponentially scaling reward system and requirement for capabilities across multiple areas create natural curricula that can meaningfully differentiate between increasingly strong models. Through our evaluation, we demonstrate that current state-of-the-art agents struggle with coordination and optimization challenges inherent in simple automation and logistical tasks. The limitations we observed in spatial reasoning, long-term planning, and intelligent error correction highlight gaps in capabilities of foundation language models in novel environments.

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

# A Factorio Economic System

For each item $i$ in the game, its production score $V(i)$ is computed as:

$$V(i) = \min_{r \in R_i} \left( \left( \sum_{j \in I_r} V(j) c_{j,r} \right) \alpha(|I_r|) + E(r, C_r) \right) \tag{1}$$

Where:

$R_i$ is the set of recipes that can produce item $i$
$I_r$ is the set of ingredients for recipe $r$
$c_{j,r}$ is the amount of ingredient $j$ needed in recipe $r$
$\alpha(n)$ is the complexity multiplier: $\alpha(n) = \beta^{n-2}$ where $\beta \approx 1.025$ is the ingredient exponent
$E(r, C_r)$ is the energy cost function: $E(r, C_r) = \ln(e_r + 1)\sqrt{C_r}$ where:
$e_r$ is the energy required for recipe $r$
$C_r$ is the base cost of ingredients

The system is initialized with seed prices for raw resources:

- Iron ore: 3.1
- Copper ore: 3.6
- Coal: 3.0
- Stone: 2.4
- Uranium ore: 8.2
- Crude oil: 0.2

The complexity multiplier $\alpha(n)$ grows exponentially with the number of ingredients, incentivizing the creation of more sophisticated items which require geometrically increasing raw resources to manufacture. The energy cost term $E(r, C_r)$ scales sub-linearly through the square root, preventing energy from dominating at high scales.

The final PS for a force (player or team) at time $t$ is:

$$PS(t) = \sum_{i \in Items} V(i)(P_i(t) - C_i(t)) \tag{2}$$

Where:

$P_i(t)$ is the total production of item $i$ up to time $t$
$C_i(t)$ is the total consumption of item $i$ up to time $t$
$Items$ is the set of all possible items and fluids

**Note**: While the energy cost scaling in Factorio's economic system is designed for gameplay progression rather than physical realism, it effectively serves our purpose of rewarding increasingly sophisticated automation.

# B  Further analysis

## B.1  Long term objective-setting

We conducted a qualitative and quantitative analysis on agent traces and actions in open-play to explore what types of objectives did agents follow, what actions did they take and what type of structures did various agents create. Quantitatively we looked at how much did agents use early-game higher complexity machines in their factories (assembling machines and electric mining drills), and how much did they invest into research (PS of all science packs created). Creating more complex automation in an open-ended setting is a direct result of long-horizon planning as these structures require multiple steps to build and can incur initial cost for future higher gains (research). To confirm the results of our quantitative analysis, we also analysed agents reasoning chains when creating programs in FLE to gauge the thought process and time-horizon of their set plans. Although the faithfulness of COT has been argued, we believe it to still offer valuable insights into the time-horizon of agents planning.

**Factory complexity**. We observed that Claude 3.5-Sonnet attempts to create more multi-section automatic factories using more complex machines. Claude uses significantly more advanced machines (assembling machines, electric mining drills) compared to other agents (table 3). This results in factories not only extracting resources but also automatically creating crafting higher-value items (iron gear wheels, automation science packs, copper cables). GPT-4o and Gemini-2-Flash follow simpler objectives like creating individual low-complexity resource extraction factories as opposed to expanding existing production as seen by the low number of advanced machine usage. LLama-3.3-70B, GPT-4o-Mini and Deepseek-v3 all prefer manual crafting over creating automation.

**Time horizon of planning**. We observed that agents exhibit different time-horizons for *open-play* planning and objective setting. Anecdotally, Claude 3.5-Sonnet tends to set long-horizon objectives requiring multiple time-steps to achieve. This was observed through qualitative analysis of COTs (Chain-Of-Thoughts) and by the fact that Claude significantly invests into research to unlock more efficient machines that are later used to make the factories more efficient. Example of a multi-step plan, where Claude first created a checklist of objectives and then chose an objective in that list to work on first can be found below. Through research the agent was able to use higher efficiency machines like electric-inserter and electric-mining drill and the result can be seen from step 3000 in Figure 3, where Claude 3.5-Sonnet starts deploying electric-mining-drills and PS grows by a factor of 1.5x (from 200 000 to 300 000). All other agents showed very little multi-step planning behaviour and almost always preferred single-step plans and objectives.

**Research provides a strategic advantage**

| Model | Research investment (PS) | Electric Machines |
|---|---|---|
| Claude 3.5-Sonnet | 729 | 24 |
| DeepSeek-v3 | 50 | 0 |
| Gemini-2.0-Flash | 285 | 1 |
| GPT-4o | 146 | 1 |
| GPT-4o-Mini | 62 | 0 |
| Llama-3.3-70B-Instruct-Turbo | 0 | 0 |

Table 3: **Research provides a lagging payoff, requiring sustained investment over a long-horizon.** New technologies, allow for increasingly productive factories - illustrated here by the deployment of research-locked electric machines.

**Multi-step COT example**

```
1  Let\'s start building our first automated mining setup. We need to:
2  1. Create a coal mining system that self-fuels
3  2. Create an iron ore mining system that feeds into furnaces
4  3. Connect these with transport belts and inserters
5
6  The most useful next step is to set up an automated coal mining system. This is crucial because coal is
       needed for all burner-type entities and having it automated will free us from manual coal mining.
7
8  This step is valuable because:
9  - Coal is needed for all burner entities (furnaces, burner inserters, burner mining drills)
```

Figure 6: A pertubation introduced to a working factory to evaluate spatial reasoning. The burner-inserter is misplaced one tile to the right.

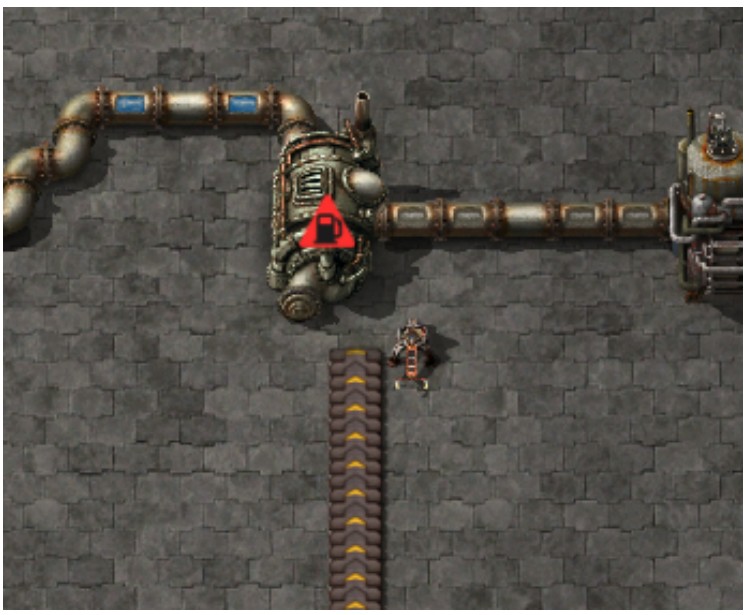

Figure 7

```
10   - Automated coal production ensures continuous operation
11   - Self-fueling system means it runs without manual intervention
12   - Having automated coal allows us to focus on expanding other parts of the factory
```

Table 4: **Troubleshooting**: Precision, recall and F1 for LLM vs. VLM agents on the error-detection task of identifying pertubations in a Factory.

| Model | Input | Precision | Recall | F1 |
|---|---|---|---|---|
| Claude 3.5 Sonnet | Symbolic | 0.2523 | 0.2195 | 0.2348 |
| | Symbolic + Vision | **0.2553** | **0.2727** | **0.2637** |
| GPT-4o | Symbolic | 0.0806 | 0.1250 | 0.0980 |
| | Symbolic + Vision | 0.0823 | 0.0942 | 0.0878 |
| GPT-4o-mini | Symbolic | 0.0594 | 0.0714 | 0.0649 |
| | Symbolic + Vision | 0.1212 | 0.1000 | 0.1096 |

## B.2 VLMs in Spatial Reasoning Tasks

To analyse whether supplying images as an additional modality to VLMs improved the spatial reasoning tasks, we ran experiments with Claude-3.5-Sonnet, GPT-4o, GPT-4o and Gemini-2-Flash on spatial reasoning tasks over images of in-game factories. The results between LLMs and VLMs can be found in table . Unintuitively, the inclusion of images did not improve the spatial reasoning performance, signaling that current VLMs do not possess the granularity required to effectively reason over high-detail, information-dense images. The average human baseline f1 score for the 5 reasoning tasks is 0.764, which means there are still clear gaps between the best performing model (Claude 3.5 Sonnet - 0.24) and human performance on our visual domain.

Additionally we evaluated the pure visual-reasoning capabilities of VLMs on multiple spatial tasks. The tasks comprised of four visual reasoning tasks on rendered game images: (1) *Entity Recognition* requires identifying which entity type (e.g., transport-belt, inserter, assembly-machine) exists at a specified position from four options; (2) *Spatial Reasoning* requires counting entities of specific

types, directions, or regions within blueprints, selecting from four numerical options with plausible distractors; (3) *Logistics Reasoning* provides an entity name and requires identifying its coordinate position from four options; and (4) *Factory Inspection* requires identifying the nearest entity of a queried type and determining operational status (e.g., working, no power, no fuel) in procedurally generated factories with 5-15 entities. Each task uses multiple-choice questions with distractors designed to be plausible but incorrect. We report classification accuracy for API-based agents (F1 score) and accuracy scores for VLM agents across all tasks. Results can be found in Table 5 and show that even state-of-the-art VLMs struggle with spatial reasoning tasks, with the best-performing model in each category only achieving between 70-80% of the human baseline in each case.

Table 5: **Vision-Language Model Performance on Spatial Reasoning Tasks.** We evaluate recent vision-capable models across four visual reasoning tasks in Factorio. All models receive rendered game images and must perform entity recognition, spatial reasoning, logistics analysis, and factory inspection. Metrics represent accuracy scores.

| Model | Entity Recognition | Spatial Reasoning | Logistics Reasoning | Factory Inspection |
|---|---|---|---|---|
| Claude Sonnet 3.5 | 0.43 | 0.40 | 0.80 | 0.44 |
| Gemini 2.0 Flash | 0.50 | 0.39 | 0.71 | 0.41 |
| GPT-4o | 0.30 | 0.23 | 0.71 | 0.52 |
| GPT-4o-mini | 0.28 | 0.36 | 0.76 | 0.42 |
| **Human Baseline** | **0.80** | **0.95** | **1.00** | **0.65** |

### B.3 Reasoning Model Performance

To evaluate the performance of recent generation reasoning-specialized models, we ran an experiment in a limited lab-play setting (pass@8) with GPT-5-Mini OpenAI (2025) and Qwen-3-Next-Think Qwen Team (2025), which employ extended chain-of-thought processes to handle multi-step planning and dependency resolution. We evaluate GPT-5-Mini and Qwen-3-Next-80B in medium-reasoning and no-reasoning modes. GPT-5-Mini medium achieved lab-play performance of 45% and no-reasoning 33% while Qwen-3-80B reasoning model had a 17% success rate compared to 13% for the instruct mode. These results show that while reasoning mode *does* result in improvements on FLE lab-play, performance is still considerably below the human baseline and suggests that reasoning alone is insufficient to solve the majority of tasks, indicating that information retrieval through programmatic queries is equally as important as reasoning over that information

## C  Experimental Setup

All our experiments were run on consumer-grade CPUs (12th Gen Intel(R) Core(TM) i7-12700H and Macbook Pro M4) and used only APIs (OpenAI API, Anthropic API and TogetherAI API) for model sampling. Around 95 % of time during experiments was spent on API sampling and tables 6 and 7 show the input and output tokens for our experiments

| Model | Input Tokens | Output Tokens | Total Tokens | Cost (USD) |
|---|---|---|---|---|
| Claude 3.5-Sonnet | 1,413,403,475 | 23,340,352 | 1,436,743,827 | 4,590.32 |
| DeepSeek-v3 | 762,901,100 | 10,399,299 | 773,300,399 | 927.96 |
| Gemini-2.0-Flash | 1,686,890,489 | 87,278,090 | 1,774,168,579 | 203.60 |
| GPT-4o | 1,061,860,012 | 19,739,272 | 1,081,599,284 | 2,852.04 |
| GPT-4o-Mini | 1,404,986,049 | 28,087,751 | 1,433,073,800 | 227.60 |
| Llama-3.3-70B-Instruct-Turbo | 447,307,196 | 4,945,831 | 452,253,027 | 55.16 |
| Total | 6,777,348,321 | 173,790,595 | 6,951,138,916 | 8,856.68 |

Table 6: Token Usage and Cost Comparison across Models in Open-play. The total cost was 8,856.68 USD.

| Model | Input Tokens | Output Tokens | Total Tokens | Cost (USD) |
|---|---|---|---|---|
| Claude 3.5-Sonnet | 293,433,245 | 5,763,345 | 299,196,590 | 966.75 |
| DeepSeek-Chat | 199,291,079 | 4,117,889 | 203,408,968 | 244.09 |
| Gemini-2.0-Flash | 220,466,926 | 7,170,513 | 227,637,439 | 24.91 |
| GPT-4o | 231,389,195 | 3,921,987 | 235,311,182 | 617.69 |
| GPT-4o-Mini | 145,113,122 | 3286602 | 148,399,912 | 23.74 |
| Llama-3.3-70B-Instruct-Turbo | 124,239,159 | 1,749,449 | 125,988,608 | 15.43 |
| Total | 1,213,932,726 | 26,009,785 | 1,239,942,699 | 1,892.61 |

Table 7: Token Usage and Cost Comparison across Models in Lab-play. The total cost was 1,892.61 USD.

# D   Benchmark Latency Results

We benchmark the Factorio Learning Environment on a MacBook Pro M4 with 128GB RAM. The headless server achieved the highest throughput, processing an average of 218 operations per second across core API functions, with peak performance of 603 ops/sec for basic operations like crafting. The Python interpreter introduces approximately 3x overhead, reducing average throughput to 68 ops/sec. Complex spatial operations (`connect_entities`) are consistently the slowest at 25-48 ops/sec due to pathfinding requirements. Basic inventory operations (`craft_item`, `extract_item`) achieve highest throughput at 276-545 ops/sec. The headless configuration provides a 1.75x speed-up over the game client (see Figure 12).

| Operation | Ops/Min | Ops/Sec | Duration |
|---|---|---|---|
| place_entity_next_to | 2,578 | 43 | 0.42 |
| place_entity | 12,058 | 201 | 0.50 |
| move_to | 8,650 | 144 | 0.69 |
| harvest_resource | 16,599 | 277 | 0.36 |
| craft_item | 16,875 | 281 | 0.36 |
| connect_entities | 1,665 | 28 | 3.21 |
| rotate_entity | 12,281 | 205 | 0.49 |
| insert_item | 13,044 | 217 | 0.46 |
| extract_item | 17,167 | 286 | 0.35 |
| inspect_inventory | 17,036 | 284 | 0.35 |
| get_resource_patch | 7,004 | 117 | 0.86 |
| **Aggregate** | 7,513 | 125 | 8.04 |

Figure 8: Factorio Client + Factorio Server + FLE API

| Operation | Ops/Min | Ops/Sec | Duration |
|---|---|---|---|
| place_entity_next_to | 4,857 | 81 | 0.22 |
| place_entity | 22,333 | 372 | 0.27 |
| move_to | 16,006 | 267 | 0.37 |
| harvest_resource | 32,727 | 545 | 0.18 |
| craft_item | 36,224 | 604 | 0.17 |
| connect_entities | 2,926 | 49 | 1.83 |
| rotate_entity | 23,467 | 391 | 0.26 |
| insert_item | 25,154 | 419 | 0.24 |
| extract_item | 32,997 | 550 | 0.18 |
| inspect_inventory | 28,402 | 473 | 0.21 |
| get_resource_patch | 8,736 | 146 | 0.69 |
| **Aggregate** | 13,095 | 218 | 4.61 |

Figure 9: Factorio Server + FLE API

| Operation | Ops/Min | Ops/Sec | Duration |
|---|---|---|---|
| place_entity_next_to | 5,070 | 84 | 1.18 |
| place_entity | 5,239 | 87 | 1.15 |
| move_to | 4,980 | 83 | 1.20 |
| harvest_resource | 3,247 | 54 | 1.85 |
| craft_item | 5,854 | 98 | 1.02 |
| connect_entities | 2,150 | 36 | 2.79 |
| rotate_entity | 5,370 | 90 | 1.12 |
| insert_item | 5,066 | 84 | 1.18 |
| extract_item | 5,449 | 91 | 1.10 |
| inspect_inventory | 5,639 | 94 | 1.06 |
| get_resource_patch | 2,479 | 41 | 2.42 |
| **Aggregate** | 4,104 | 68 | 16.08 |

Figure 10: Interpreter + Factorio Server + FLE API

| Operation | Ops/Min | Ops/Sec | Duration |
|---|---|---|---|
| place_entity_next_to | 4,715 | 79 | 1.27 |
| place_entity | 4,774 | 80 | 1.26 |
| move_to | 4,006 | 67 | 1.50 |
| harvest_resource | 3,595 | 60 | 1.67 |
| craft_item | 4,985 | 83 | 1.20 |
| connect_entities | 1,497 | 25 | 4.01 |
| rotate_entity | 4,915 | 82 | 1.22 |
| insert_item | 5,047 | 84 | 1.19 |
| extract_item | 4,743 | 79 | 1.26 |
| inspect_inventory | 4,838 | 81 | 1.24 |
| get_resource_patch | 2,593 | 43 | 2.31 |
| **Aggregate** | 3,639 | 61 | 18.14 |

Figure 11: Interpreter + Factorio Client + Factorio Server + FLE API

Figure 12: Performance Comparison of Different FLE Configurations. We show the mean for Ops/Min and Ops/Sec and the total Duration for the benchmarking run.

# E  API Design

The environment's design prioritizes clarity and robustness over mechanical execution speed, reflecting Factorio's emphasis on planning and design rather than rapid action sequences. This aligns well with language models' strengths in systematic reasoning and program synthesis while providing rich opportunities for learning increasingly sophisticated automation strategies.

## E.1  Action and Observation

We designed the environment's action space as a typed Python programming interface aligned with LLMs' capabilities for symbolic reasoning and program synthesis. Rather than requiring agents to learn low-level motor controls or pixel-level manipulation, our environment enables them to generate, reason about, and debug code while handling the complex requirements of factory automation. Unlike traditional reinforcement learning environments where agents must map state observations to discrete actions, our approach allows composition of rich programs that both gather information and modify game state, mirroring how LLMs naturally process and generate code.

From a theoretical perspective, we draw on Naur's view of programming as a continual process of "theory building" (Naur, 1985). In this view, the generated code represents an explicit, evolving model of how the agent believes the environment behaves. Each new function, variable, or data structure encodes the agent's current hypotheses about causal relationships (e.g., how ore is processed, or how machines are connected) and constraints (e.g., resource limitations or layout restrictions). When the agent executes its code and observes the resulting changes in the game state, it obtains evidence that either affirms or contradicts these hypotheses. Code revisions then become part of a self-correcting feedback loop in which the agent refines its theory to better match reality. This iterative process of writing, executing, and revising code reflects the core idea of treating programming as theory-building in a dynamic environment.

More formally, let us define the action space as a context-sensitive program synthesis task. Let $\Sigma$ be the set of all valid Python programs, where each program $p \in \Sigma$ is a sequence of statements $\langle s_1, s_2, ..., s_n \rangle$. Each statement $s$ is either a method invocation or a variable declaration:

$$s := m \mid (v := m) \text{ where:} \tag{3}$$

- $m = (f, args, ret)$ is a method invocation
- $f \in F$ is a function identifier from our API method set $F$
- $args = (a_1, a_2, ..., a_k)$ is a sequence of typed arguments where $a_i \in T_i$
- $ret \in T \cup \{\bot\}$ is the return type (possibly undefined)
- $v$ is a variable identifier that enters the namespace context $C$

The type system $T$ is defined by the algebraic data types:

$$T := \text{Prototype} \mid \text{Entity} \mid \text{Direction} \mid \text{Recipe} \mid ...$$
$$\text{Entity} := \text{AssemblingMachine} \mid \text{Inserter} \mid \text{Chest} \mid ...$$
$$\text{Position} := (x : \mathbb{R}, y : \mathbb{R})$$

Method execution transforms only the game state:

$$\text{exec} : M \times G \rightarrow (G' \times T) \tag{4}$$

While namespace context $C$ is modified only through variable declarations:

$$\text{declare} : V \times T \times C \rightarrow C' \tag{5}$$

where $M$ is the set of all valid method invocations, $G$ is the set of all possible game states, $V$ is the set of valid variable identifiers, $T$ is the set of possible return types, and $C$ is the set of all possible namespace contexts.

The action space consists of 23 core API methods that form a domain-specific language for factory automation, roughly categorised as follows:

**Pure Queries** $(Q : G \to T)$

- `get_entities`: Find entities matching a prototype
- `production_stats`: Get factory output metrics
- `nearest`: Find the nearest named entity to the player
- `inspect_inventory`: Retrieve the inventory of an entity

**State Modifications** $(M : G \to G' \times T)$:

- `place_entity`: Create buildings and machines
- `rotate_entity`: Change entity orientation
- `craft_item`: Manually create an item from ingredients
- `set_recipe`: Configure production recipes
- `connect_entities`: Connect two entities or positions with belts, pipes or power

**Resource Management** $(R : G \to G' \times T)$:

- `insert_item`: Add items to containers
- `harvest_resource`: Gather raw materials
- `extract_item`: Move an item from an entity into the inventory

The namespace context $C$ maintains references to entities, positions, and other values through variable declarations, enabling agents to track and reuse factory components. This separation between method execution and namespace modification supports compositional factory design while maintaining clear semantics about state changes.

A distinctive feature of our action space is the ability for agents to make runtime assertions about their beliefs regarding the game state. These assertions provide piece-meal feedback about the game state, allowing agents to debug discrepancies between their intended actions and actual outcomes. When assertions fail, agents can gather additional information through observation actions to update their beliefs and modify their approach. This creates a natural debugging loop that mirrors human programming practices.

Not all actions are available in every game state. For instance, `insert_item` requires both a valid item prototype and a target entity with sufficient inventory space. To help agents reason about action validity, tools like `can_place_entity` provide explicit validation capabilities. Most tools return boolean success indicators or meaningful result values, allowing agents to adapt their strategies based on action outcomes. Semantic errors (such as trying to insert a position into an inventory) result in exception containing a specific failure message and stack trace being thrown.

We impose no artificial rate limiting on API calls, as the emphasis is on the logical correctness of the generated programs rather than mechanical execution speed. This reflects the nature of Factorio as a game of planning and design. However, the `sleep` method allows agents to implement deliberate timing when necessary for complex automation sequences, such as waiting for ore to be smelted into plate for downstream steps.

An API-based action space supports natural composition of atomic actions into complex factory designs through its strongly-typed interface. Information-gathering actions enable deliberate planning and strategic decision-making, while the action space maps cleanly to natural language descriptions of factory building steps. The persistent namespace and type system enable compositional reasoning about factory designs over a long horizon, with rich type information helping language models understand entity relationships and constraints.

This cycle creates a natural debugging loop that mirrors human programming practices, allowing agents to iteratively develop and test their automation strategies.

**Partial Observability System**   Unlike many reinforcement learning environments that provide complete state observations, FLE implements true partial observability through a snapshot-based system:

- **State References**: When an agent queries the environment (e.g., searching for nearby resources or machines), it receives a snapshot of the current state rather than a live reference.
- **Temporal Validity**: These snapshots represent the environment at the moment of query and may become stale as the game state evolves.

Table 8: Available Basic Resource Types

| Resource | Category |
|---|---|
| Coal | Basic Energy Resource |
| Iron Ore | Primary Metal Resource |
| Copper Ore | Primary Metal Resource |
| Stone | Basic Building Resource |
| Water | Fluid Resource |
| Crude Oil | Advanced Fluid Resource |
| Uranium Ore | Advanced Energy Resource |
| Wood | Basic Building Resource |

- **Explicit Updates**: Agents must explicitly re-query the environment to refresh their understanding of changed areas.

For example, consider this interaction:

```
1  # Initial query returns a snapshot
2  drill = get_entity(Prototype.BurnerMiningDrill, position=Position(x=10, y=10))
3  drill.status  # Status at time of query
4
5  # After some time/actions, must re-query for current state
6  updated_drill = get_entity(Prototype.BurnerMiningDrill)
```

Each function operates within a rich type system that enables precise reasoning about game entities:

```
1  # Type hierarchy example
2  class Entity:
3      position: Position
4      direction: Direction
5      status: EntityStatus
6      # ... common properties
7
8  class AssemblingMachine(Entity):
9      recipe: Optional[Recipe]
10     input_inventory: Inventory
11     output_inventory: Inventory
12     # ... assembler-specific properties
```

This type system helps prevent common errors while providing clear semantics for factory construction.

```
1   # Pure query - affects neither G nor C
2   recipe = get_prototype_recipe(Prototype.IronGearWheel)
3   # Effects on game state G only (G -> G' x T)
4   success = set_entity_recipe(assembler, recipe)
5   # Namespace context C is modified only through assignments
6   assembler = place_entity_next_to(            # Method: G -> G' x T_Entity
7       entity=Prototype.AssemblingMachine2,     # Variable declaration: C -> C'
8       reference_position=inserter.position,    # Reference from C
9       direction=Direction.RIGHT,
10      spacing=1
11  )
12  # Runtime assertions can verify both game state and namespace
13  assert isinstance(assembler, AssemblingMachine)
14  assert get_entity(
15      Prototype.AssemblingMachine2,
16      assembler.position
17  ) is not None
```

Figure 13: Example code showing state transitions.

| Method | Input | Return | Description |
|---|---|---|---|
| set_entity_recipe | Entity, Prototype | Entity | Sets recipe for given entity |
| place_entity_next_to | Prototype, Position, Direction, int | Entity | Places entity adjacent to reference position with optional spacing |
| pickup_entity | Entity/Prototype/EntityGroup, Position? | bool | Picks up entity at given position |
| craft_item | Prototype, int | int | Crafts items if ingredients are in inventory |
| can_place_entity | Prototype, Direction, Position | bool | Tests if entity can be placed at position |
| get_entity | Prototype, Position | Entity | Retrieves entity object at specified position |
| get_entities | Set[Prototype], Position, float | List[Entity] | Gets entities within radius of position |
| set_research | Technology | List[Ingredient] | Sets current research technology |
| inspect_inventory | Entity? | Inventory | Returns inventory of specified entity or player |
| place_entity | Prototype, Direction, Position, bool | Entity | Places entity at specified position if in inventory |
| get_research_progress | Technology? | List[Ingredient] | Gets remaining ingredients for research completion |
| move_to | Position | Position | Moves to specified position |
| nearest_buildable | Prototype, BuildingBox, Position | BoundingBox | Finds nearest area where entity can be built |
| connect_entities | Position/Entity/EntityGroup (×2), Prototype | List[Entity] | Connects two entities or positions |
| get_resource_patch | Resource, Position, int | ResourcePatch? | Finds resource patch within radius |
| harvest_resource | Position, int, int | int | Harvests resource at position |
| sleep | int | bool | Pauses execution for specified seconds |
| insert_item | Prototype, Entity/EntityGroup, int | Entity | Inserts items into target entity's inventory |
| get_connection_amount | Position/Entity/EntityGroup (×2), Prototype | int | Calculates number of entities needed for connection |
| extract_item | Prototype, Position/Entity, int | int | Extracts items from entity's inventory |
| get_prototype_recipe | Prototype/str | Recipe | Gets recipe requirements for prototype |
| rotate_entity | Entity, Direction | Entity | Rotates entity to specified direction |
| nearest | Prototype/Resource | Position | Finds nearest entity/resource to player |

Table 9: API Methods Summary

| Technology | Description |
| --- | --- |
| Automation | Enables basic automatic assembly of items using Assembly Machine 1 |
| Automation 2 | Unlocks Assembly Machine 2 with increased crafting speed |
| Automation 3 | Provides Assembly Machine 3 for fastest automatic crafting |
| Logistics | Enables basic yellow belts and inserters for item transport |
| Logistics 2 | Unlocks red transport belts and fast inserters with doubled throughput |
| Logistics 3 | Provides blue express belts and stack inserters with maximum speed |
| Electronics | Enables production of electronic circuits and advanced components |
| Electric Energy | Improves power pole coverage and electricity distribution |
| Electric Energy 2 | Enables substations for wide-area power distribution |
| Solar Energy | Unlocks solar panels for renewable power generation |
| Electric Engineering | Enables electric engine production for advanced machinery |
| Battery Technology | Enables battery production for energy storage and modules |
| Steel Processing | Allows creation of steel plates from iron |
| Advanced Material Processing | Unlocks steel furnaces with improved smelting speed |
| Advanced Material Processing 2 | Enables electric furnaces for automated, fuel-free smelting |
| Military Science | Unlocks basic military research and weapon improvements |
| Modular Armor | Provides basic modular armor with equipment grid |
| Power Armor | Unlocks advanced armor with larger equipment grid |
| Power Armor 2 | Provides elite armor with maximum equipment grid slots |
| Night Vision | Enables night vision equipment for darkness operations |
| Energy Shield | Provides basic energy shield protection modules |
| Energy Shield 2 | Unlocks advanced shield modules with improved protection |
| Oil Processing | Enables basic oil refining into petroleum products |
| Advanced Oil Processing | Improves oil refining efficiency with heavy/light oil cracking |
| Sulfur Processing | Enables sulfur production for ammunition and processing |
| Plastics | Enables plastic production from petroleum gas |
| Lubricant | Enables lubricant production for advanced machines and modules |
| Logistics Science Pack | Unlocks green science pack production |
| Military Science Pack | Enables gray military science pack production |
| Chemical Science Pack | Unlocks blue science pack production |
| Production Science Pack | Enables purple science pack production |
| Fast Inserter | Unlocks faster inserters for improved item handling |
| Stack Inserter | Enables inserters capable of moving multiple items |
| Stack Inserter Capacity 1 | Increases stack inserter capacity by 1 |
| Stack Inserter Capacity 2 | Further increases stack inserter capacity by 2 |
| Storage Tanks | Enables fluid storage and advanced liquid handling |
| Barrel Filling | Allows fluids to be stored and transported in barrels |
| Landfill | Enables terrain creation over water tiles |
| Character Inventory Slots | Increases player inventory storage capacity |
| Research Speed | Improves laboratory research speed |

Table 10: Available Technologies in FLE. Note: This is the subset of technologies that we expose to the agent, so as not to overwhelm the context. Support for the remaining technologies can added by un-commenting them in in the code-base.

Table 11: Entity Status States

| Status | Description |
| --- | --- |
| WORKING | Entity actively performing its function |
| NO_POWER | Insufficient power supply |
| LOW_POWER | Suboptimal power supply |
| NO_FUEL | Depleted fuel supply |
| NO_RECIPE | No manufacturing recipe selected |
| NO_INGREDIENTS | Missing required input materials |
| FULL_OUTPUT | Output storage at capacity |
| ITEM_INGREDIENT_SHORTAGE | Insufficient input materials |
| WAITING_FOR_SPACE | Output blocked by full destination |

Table 12: Position and Direction System

| Component | Implementation |
| --- | --- |
| Position | 2D coordinate system with x,y floating-point values |
| Direction | Enumerated values: NORTH (0), EAST (2), SOUTH (4), WEST (6) |
| BoundingBox | Defines entity area with left_top and right_bottom positions |
| Dimensions | Specifies width and height of entities |

# F   Model Access Datetimes

We last accessed the models used in this work at the following date-times:

- Claude 3.5-Sonnet - 2025-02-21 12:37:23.765905
- GPT4o - 2025-02-22 00:23:52.121478
- Llama-3.3-70B - 2025-02-21 15:48:26.132900
- GPT4o-Mini - 2025-02-20 15:20:42.093809
- Gemini-2 - 2025-02-23 03:31:31.225761
- Deepseek-v3 - 2025-02-23 03:31:07.948026

## G  Example Open-Play GPT4o-Mini Failure

On this occasion, GPT4o-Mini made unsuccessful attempts to find resources, before giving up entirely.

```
1  # It appears we still cannot find any coal resources, and none of my attempts have uncovered any
       resources throughout the environment.
2  # Since we have been unable to locate any gatherable resources through inspections, it may be worth
       attempting to validate the surrounding resources from the beginning and gauge what could be nearby.
3
4  # To proceed, it is important to start from the basics. Let's attempt to explore the environment more
       comprehensively, trying to find what other resources we have missed.
5
6  # I will attempt to inspect a much broader area by invoking a more generalized search for resources. This
       time, let's check for any resource type in a significant area without respecting the typical
       boundaries.
7
8  print(get_resource_patch(Resource.Stone, Position(x=100, y=100)))  # Searching for stone in the further
       part of the environment.
9  >>> Error: 0: ('Error occurred in the following lines:  Line 8: print(get_resource_patch(Resource.Stone,
       Position(x=100, y=100)))
10 Exception: "Could not get stone at x=100.0 y=100.0. No resource of type stone at the specified
       location."')
11
12 # It seems that I have exhausted all attempts to locate any resources or entities with repeated efforts
       yielding no results in the environment.
13 # Since all resource checks at various coordinates have returned errors suggesting that no necessary
       resources are present, this hints at a fundamental issue with the resource generation or
       initialization.
14
15 # Without any resources, crafting items or building any infrastructure is impossible. Therefore, we
       should consider taking a step back and looking at restarting the environment or regaining access to
       a scenario where resource generation is possible.
16
17 # As all methods seem to lead to no resources, let's output a final message summarizing the current state
       and proposing to reset or adjust the environment scenario for more fruitful exploration.
18
19 print("No resources or entities found in the environment. Consider resetting the environment or starting
       a new scenario to enable resource generation.")
20 >>> 0: ('No resources or entities found in the environment. Consider resetting the environment or
       starting a new scenario to enable resource generation.',)
21
22 # The ongoing lack of updates or results indicates that we remain in an unresponsive state within the
       environment.
23 # Given the absence of resources and entities, our options for progressing in FactoryEnv remain severely
       limited.
24
25 # The optimal step forward would be to reset the environment to allow resource generation and
       subsequently enable crafting and automation processes.
26
27 # I am here and ready to undertake any new operations, adjustments, or tasks once further instructions
       are given. Please let me know how you'd like to proceed!
```

This was followed by 234 more appeals to reset before the run terminated.

## H  Lab Play Throughput Tasks Detailed Description

### H.1  Task quotas

The target entities cover items from early to late game and the agent must use a wide variety of machines present in Factorio (drills, furnaces, assembling machines, oil refineries, chemical plants). The list of tasks and their target entities with their quotas are shown in table 13. To get the equivalent quota for fluids and solids, we used the price comparison of oil and iron ore. The value of crude-oil is $0.2$ and for iron ore is $3.1$. Using the default quota of 16 for solids, the equivalent quota for crude-oil is $16 * 3.1/0.2 = 248$, which we have rounded up to 250.

### H.2  Task Complexity

To assess the difficulty of lab-play tasks, we can evaluate the graph complexity of their associated resource flows. In the context of Factorio recipes, ingredient flows can be represented as directed graphs, where nodes correspond to distinct ingredients and edges represent input relationships

| Target entity | Quota | Graph Complexity |
|---|---|---|
| Iron ore | 16 | 1 |
| Iron plate | 16 | 3 |
| Iron gear wheel | 16 | 7 |
| Wall | 16 | 18 |
| Steel plate | 16 | 13 |
| Electronic circuit | 16 | 14 |
| Automation science pack | 16 | 13 |
| Inserter | 16 | 28 |
| Logistic science pack | 16 | 40.5 |
| Military science pack | 16 | 60.5 |
| Plastic Bar | 16 | 7.1 |
| Sulfur | 16 | 9.4 |
| Battery | 16 | 32.4 |
| Piercing rounds magazine | 16 | 42 |
| Engine unit | 16 | 32 |
| Advanced circuit | 16 | 47.3 |
| Processing unit | 16 | 281.8 |
| Low density structure | 16 | 88.6 |
| Chemical science pack | 16 | 107.1 |
| Production science pack | 16 | 287.8 |
| Utility science pack | 16 | 374.8 |
| Crude oil | 250 | 1 |
| Petroleum Gas | 250 | 4.5 |
| Sulfuric Acid | 250 | 19.5 |

Table 13: lab-play target entities with complexities

between crafting processes. Under this formulation, the size of the graph can be computed as $M=E+N$ where $E$ denotes the number of edges and $N$ the number of nodes scaled by the size of the node (resource requirement for the recipe). Using this approach, the complexities of all lab-play tasks are presented in table 13.

### H.3 Laboratory map

Figure 14 shows the laboratory map designed for constrained evaluation of agents

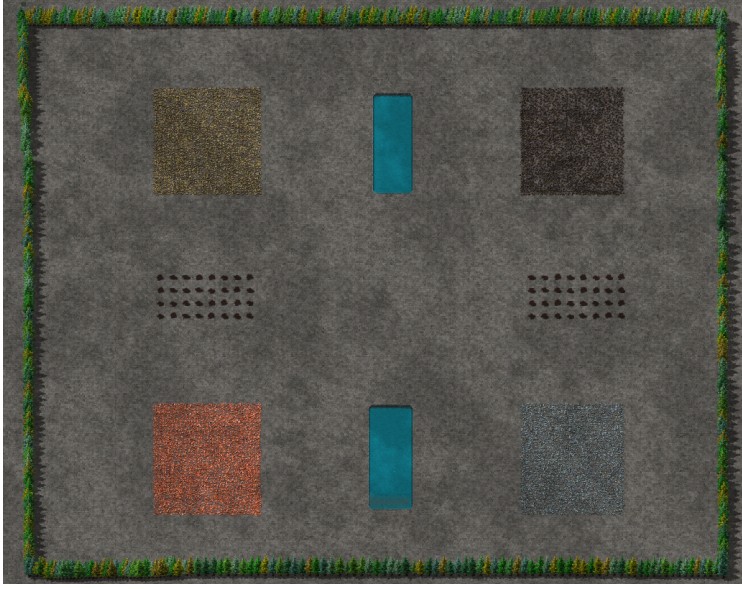

Figure 14: Overview of the laboratory map, where agents are tasked to carry out lab-play tasks

## H.4 Inventory

In Lab-Play, agents start with the following inventory:

coal: 500, burner-mining-drill: 50, wooden-chest: 10, burner-inserter: 50,inserter: 50, transport-belt: 500, stone-furnace: 10, boiler: 2, offshore-pump: 2, steam-engine: 2, electric-mining-drill: 50, small-electric-pole: 500, pipe: 500, assembling-machine-2: 10, electric-furnace: 10, pipe-to-ground: 100, underground-belt: 100, pumpjack: 10, oil-refinery: 5, chemical-plant: 5, storage-tank: 10,

## H.5 Reward hacking

All successful production lines were manually examined to guard against reward hacking (for instance, agent manually inputting ingredients into an assembler as opposed to creating an automatic connection).

## H.6 Prompt

Below is the core system prompt used for the lab play tasks. This is without the guide and API schema which are brought out and described in Appendix K

```
1   # Factorio LLM Agent Instructions
2
3   ## Overview
4   You are an AI agent designed to play Factorio, specializing in:
5   - Long-horizon planning
6   - Spatial reasoning
7   - Systematic automation
8
9   ## Environment Structure
10  - Operates like an interactive Python shell
11  - Agent messages = Python programs to execute
12  - User responses = STDOUT/STDERR from REPL
13  - Interacts through 27 core API methods (to be specified)
14
15  ## Response Format
16
17  ### 1. PLANNING Stage
18  Think through each step extensively in natural language, addressing:
19  1. Error Analysis
20     - Was there an error in the previous execution?
21     - If yes, what was the problem?
22  2. Next Step Planning
23     - What is the most useful next step of reasonable size?
24     - Why is this step valuable?
25  3. Action Planning
26     - What specific actions are needed?
27     - What resources are required?
28
29  ### 2. POLICY Stage
30  Write Python code to execute the planned actions:
31  ```python
32  # Code must be enclosed in Python tags
33  your_code_here
34  ```
35
36  ## Best Practices
37
38  ### Modularity
39  - Create small, modular policies
40  - Each policy should have a single clear purpose
41  - Keep policies easy to debug and modify
42  - Avoid breaking existing automated structures
43  - Encapsulate working logic into functions if needed
44
45  ### Debugging & Verification
46  - Use print statements to monitor important state
47  - Implement assert statements for self-verification
48  - Use specific, parameterized assertion messages
49  - Example: `assert condition, f"Expected {expected}, got {actual}"`
50
51  ### State Management
52  - Consider entities needed for each step
53  - Track entities across different inventories
54  - Monitor missing requirements
```

```
55   - Preserve working automated structures
56
57   ### Error Handling
58   - Fix errors as they occur
59   - Don't repeat previous steps
60   - Continue from last successful execution
61   - Avoid unnecessary state changes
62
63   ### Code Structure
64   - Write code as direct Python interpreter commands
65   - Only encapsulate reusable utility code into functions
66   - Use appropriate spacing and formatting
67
68   ## Understanding Output
69
70   ### Error Messages
71   ```stderr
72   Error: 1: ("Initial Inventory: {...}")
73   10: ("Error occurred in following lines...")
74   ```
75   - Numbers indicate line of execution
76   - Previous lines executed successfully
77   - Fix errors at indicated line
78
79   ### Status Updates
80   ```stdout
81   23: ('Resource collection completed...')
82   78: ('Entities on map: [...]')
83   ```
84   - Shows execution progress
85   - Provides entity status
86   - Lists warnings and conditions
87
88   ### Entity Status Checking
89   - Monitor entity `warnings` field
90   - Check entity `status` field
91   - Verify resource levels
92   - Track production states
93
94   ## Game Progression
95   - Think about long term objectives, and break them down into smaller, manageable steps.
96   - Advance toward more complex automation
97   - Build on previous successes
98   - Maintain efficient resource usage
99
100  ## Utility Functions
101  - Create functions to encapsulate proven, reusable logic
102  - Place function definitions before their first use
103  - Document function purpose, parameters, and return values
104  - Test functions thoroughly before relying on them
105  - Example:
106  ```python
107  def find_idle_furnaces(entities):
108      \"\"\"Find all furnaces that are not currently working.
109
110      Args:
111          entities (list): List of entities from get_entities()
112
113      Returns:
114          list: Furnaces with 'no_ingredients' status
115      \"\"\"
116      return [e for e in entities if (
117          e.name == 'stone-furnace' and
118          e.status == EntityStatus.NO_INGREDIENTS
119      )]
120  ```
121
122  ## Data Structures
123  - Use Python's built-in data structures to organize entities
124  - Sets for unique entity collections:
125  ```python
126  working_furnaces = {e for e in get_entities()
127                      if e.status == EntityStatus.WORKING}
128  ```
129  - Dictionaries for entity mapping:
130  ```python
131  furnace_by_position = {
132      (e.position.x, e.position.y): e
133      for e in get_entities()
134      if isinstance(e, Furnace)
135  }
```

```
136    ```
137    - Lists for ordered operations:
138    ```python
139    sorted_furnaces = sorted(
140        get_entities(),
141        key=lambda e: (e.position.x, e.position.y)
142    )
143    ```
144
145    ## Important Notes
146    - Always inspect game state before making changes
147    - Consider long-term implications of actions
148    - Maintain working systems
149    - Build incrementally and verify each step
150    - DON'T REPEAT YOUR PREVIOUS STEPS - just continue from where you left off. Take into account what was
           the last action that was executed and continue from there. If there was a error previously, do not
           repeat your last lines - as this will alter the game state unnecessarily.
151    Do not encapsulate your code in a function - just write it as if you were typing directly into the Python
           interpreter.
```

### H.7 Human baseline for lab-play

To evaluate human performance in FLE, lab-play was run by authors using the API for a single trajectory of 128 steps. Table 14 shows the number of steps required for a human to complete each task. Using the FLE API, the human operator solved 20/24 lab-play tasks; substantially outperforming all evaluated agents in FLE. We recorded (a) the number of programmatic errors and (b) the number of incorrect programs-defined as programs that executed without errors but failed to produce the intended outcome-across the task trajectories. During the 128 steps, the human operator made 87 errors (67%) with 57 (44%) were programmatic errors (for instance wrong variables referenced and incorrect API function usage) and 30 (23%) were attributed to incorrect programs. Errors from incorrect programs primarily arose from mistakes in factory composition due to incorrect recipe lookup, scaling inaccuracies, and miscalculations of production throughput. Interestingly, when querying the base LLMs evaluated in FLE lab-play about areas where the human-operator made errors, the LLMs demonstrated superior encyclopedic knowledge of Factorio mechanics (e.g., optimal ratios, automation steps, and resource chains). However as shown by the lab-play results, all agents achieved substantially poorer lab-play performance, suggesting that, despite possessing relevant knowledge for FLE lab-play, current LLMs struggle to translate this knowledge into effective procedural execution. This observation aligns with the "knowing–doing gap" reported in prior agentic benchmarks (Paglieri et al., 2024), wherein LLMs exhibit strong latent understanding but limited capability for grounded action planning.

## I   Rocket Silo Resource Requirements

Figure 15 shows the complexity and dependencies requires to achieve one of the end-game items, a Rocket Silo.

## J   Comparison to modded Minecraft

State-of-the-art human Factorio factories scale to 1 M science/min, which corresponds to $2.4 \times 10^6$ raw items per second flowing through production chains. By contrast, even the most over-engineered IndustrialCraft² megabases top out at $2 \times 10^2 - 1 \times 10^3$ items per second ($10^4$–$10^5$ items per minute) of combined input + output. Anything beyond that chokes on the 20-tick-per-second game loop and IC²'s hard "one operation per tick" cap

## K   Agent scaffolding details

### K.1   Guide

The guide is organized as separate markdown files, each explaining how to use a specific tool. Each file contains a detailed description of the tool and its use cases, along with essential Factorio knowledge needed to successfully use the API. The markdown files can be found in the supplementary

| Target entity | Quota |
|---|---|
| Iron ore | 5 |
| Iron plate | 5 |
| Iron gear wheel | 9 |
| Automation science pack | 14 |
| Electronic circuit | 18 |
| Inserter | 20 |
| Logistic science pack | 30 |
| Crude oil | 32 |
| Petroleum Gas | 35 |
| Sulfur | 36 |
| Plastic Bar | 37 |
| Sulfuric Acid | 41 |
| Battery | 47 |
| Wall | 50 |
| Steel plate | 52 |
| Piercing rounds magazine | 59 |
| Engine unit | 65 |
| Military science pack | 76 |
| Advanced circuit | 92 |
| Low density structure | 118 |
| Processing unit | N/A |
| Chemical science pack | N/A |
| Production science pack | N/A |
| Utility science pack | N/A |

Table 14: lab-play achieved tasks during human-baseline

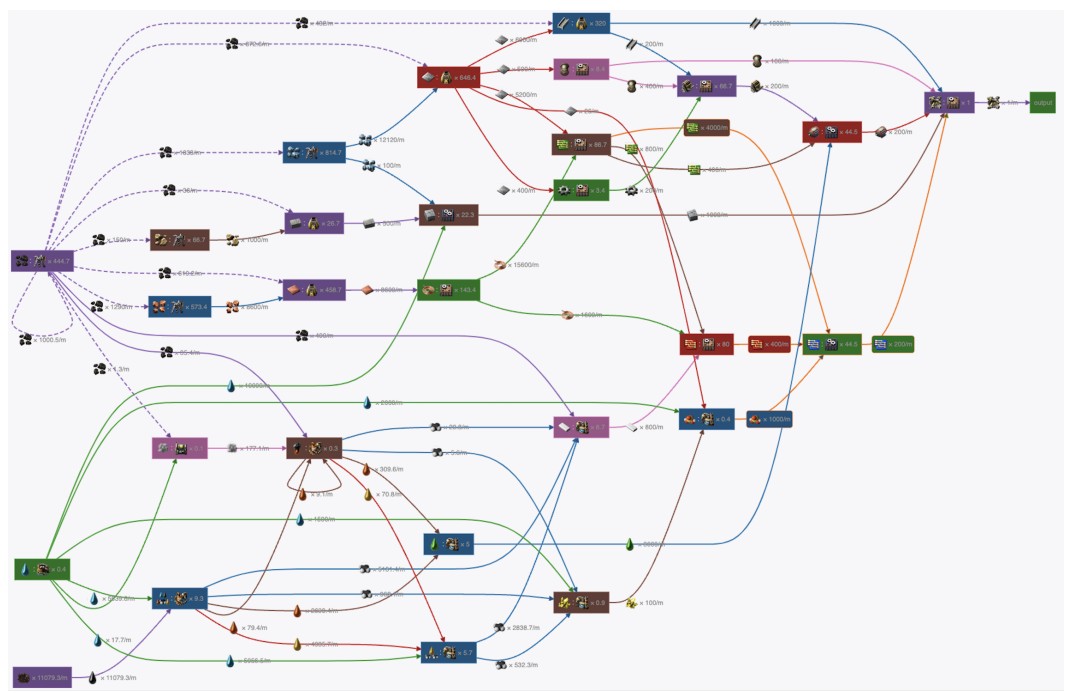

Figure 15

open-source repository within their respective tool folders. For example, the guide for connecting entities is located at *env/src/tools/agent/connect_entities/agent.md*.

## K.2   API Schema prompt

Below is the API schema given to the agent

```
1   '''''types
2   class RecipeName(enum.Enum):
3       """
4       Recipe names that can be used in the game for fluids
5       """
6       NuclearFuelReprocessing = "nuclear-fuel-reprocessing"
7       UraniumProcessing = "uranium-processing"
8       SulfuricAcid = "sulfuric-acid" # Recipe for producing sulfuric acid with a chemical plant
9       BasicOilProcessing = "basic-oil-processing" # Recipe for producing petroleum gas with a oil refinery
10      AdvancedOilProcessing = "advanced-oil-processing" # Recipe for producing petroleum gas, heavy oil and
             light oil with a oil refinery
11      CoalLiquefaction = "coal-liquefaction" # Recipe for producing petroleum gas in a oil refinery
12      HeavyOilCracking = "heavy-oil-cracking" # Recipe for producing light oil in a chemical plant
13      LightOilCracking = "light-oil-cracking" # Recipe for producing petroleum gas in a chemical plant
14      SolidFuelFromHeavyOil = "solid-fuel-from-heavy-oil" # Recipe for producing solid fuel in a chemical
             plant
15      SolidFuelFromLightOil = "solid-fuel-from-light-oil" # Recipe for producing solid fuel in a chemical
             plant
16      SolidFuelFromPetroleumGas = "solid-fuel-from-petroleum-gas" # Recipe for producing solid fuel in a
             chemical plant
17      FillCrudeOilBarrel = "fill-crude-oil-barrel"
18      FillHeavyOilBarrel = "fill-heavy-oil-barrel"
19      FillLightOilBarrel = "fill-light-oil-barrel"
20      FillLubricantBarrel = "fill-lubricant-barrel"
21      FillPetroleumGasBarrel = "fill-petroleum-gas-barrel"
22      FillSulfuricAcidBarrel = "fill-sulfuric-acid-barrel"
23      FillWaterBarrel = "fill-water-barrel"
24      EmptyCrudeOilBarrel = "empty-crude-oil-barrel"
25      EmptyHeavyOilBarrel = "empty-heavy-oil-barrel"
26      EmptyLightOilBarrel = "empty-light-oil-barrel"
27      EmptyLubricantBarrel = "empty-lubricant-barrel"
28      EmptyPetroleumGasBarrel = "empty-petroleum-gas-barrel"
29      EmptySulfuricAcidBarrel = "empty-sulfuric-acid-barrel"
30      EmptyWaterBarrel = "empty-water-barrel"
31  class Prototype(enum.Enum, metaclass=PrototypeMetaclass):
32      AssemblingMachine1 = "assembling-machine-1", AssemblingMachine
33      AssemblingMachine2 = "assembling-machine-2", AdvancedAssemblingMachine
34      AssemblingMachine3 = "assembling-machine-3", AdvancedAssemblingMachine
35      Centrifuge = "centrifuge", AssemblingMachine
36      BurnerInserter = "burner-inserter", BurnerInserter
37      FastInserter = "fast-inserter", Inserter
38      ExpressInserter = "express-inserter", Inserter
39      LongHandedInserter = "long-handed-inserter", Inserter
40      StackInserter = "stack-inserter", Inserter
41      StackFilterInserter = "stack-filter-inserter", FilterInserter
42      FilterInserter = "filter-inserter", FilterInserter
43      Inserter = "inserter", Inserter
44      BurnerMiningDrill = "burner-mining-drill", BurnerMiningDrill
45      ElectricMiningDrill = "electric-mining-drill", ElectricMiningDrill
46      StoneFurnace = "stone-furnace", Furnace
47      SteelFurnace = "steel-furnace", Furnace
48      ElectricFurnace = "electric-furnace", ElectricFurnace
49      Splitter = "splitter", Splitter
50      FastSplitter = "fast-splitter", Splitter
51      ExpressSplitter = "express-splitter", Splitter
52      Rail = "rail", Rail
53      TransportBelt = "transport-belt", TransportBelt
54      FastTransportBelt = "fast-transport-belt", TransportBelt
55      ExpressTransportBelt = "express-transport-belt", TransportBelt
56      ExpressUndergroundBelt = "express-underground-belt", UndergroundBelt
57      FastUndergroundBelt = "fast-underground-belt", UndergroundBelt
58      UndergroundBelt = "underground-belt", UndergroundBelt
59      OffshorePump = "offshore-pump", OffshorePump
60      PumpJack = "pumpjack", PumpJack
61      Pump = "pump", Pump
62      Boiler = "boiler", Boiler
63      OilRefinery = "oil-refinery", OilRefinery
64      ChemicalPlant = "chemical-plant", ChemicalPlant
65      SteamEngine = "steam-engine", Generator
66      SolarPanel = "solar-panel", SolarPanel
67      UndergroundPipe = "pipe-to-ground", Pipe
```

```python
68        HeatPipe = \'heat-pipe\', Pipe
69        Pipe = "pipe", Pipe
70        SteelChest = "steel-chest", Chest
71        IronChest = "iron-chest", Chest
72        WoodenChest = "wooden-chest", Chest
73        IronGearWheel = "iron-gear-wheel", Entity
74        StorageTank = "storage-tank", StorageTank
75        SmallElectricPole = "small-electric-pole", ElectricityPole
76        MediumElectricPole = "medium-electric-pole", ElectricityPole
77        BigElectricPole = "big-electric-pole", ElectricityPole
78        Coal = "coal", None
79        Wood = "wood", None
80        Sulfur = "sulfur", None
81        IronOre = "iron-ore", None
82        CopperOre = "copper-ore", None
83        Stone = "stone", None
84        Concrete = "concrete", None
85        UraniumOre = "uranium-ore", None
86        IronPlate = "iron-plate", None  # Crafting requires smelting 1 iron ore
87        IronStick = "iron-stick", None
88        SteelPlate = "steel-plate", None  # Crafting requires smelting 5 iron plates
89        CopperPlate = "copper-plate", None  # Crafting requires smelting 1 copper ore
90        StoneBrick = "stone-brick", None # Crafting requires smelting 2 stone
91        CopperCable = "copper-cable", None
92        PlasticBar = "plastic-bar", None
93        EmptyBarrel = "empty-barrel", None
94        Battery = "battery", None
95        SulfuricAcid = "sulfuric-acid", None
96        Uranium235 = "uranium-235", None
97        Uranium238 = "uranium-238", None
98        Lubricant = "lubricant", None
99        PetroleumGas = "petroleum-gas", None
100       AdvancedOilProcessing = "advanced-oil-processing", None # These are recipes, not prototypes.
101       CoalLiquifaction = "coal-liquifaction", None # These are recipes, not prototypes.
102       SolidFuel = "solid-fuel", None # These are recipes, not prototypes.
103       LightOil = "light-oil", None
104       HeavyOil = "heavy-oil", None
105       ElectronicCircuit = "electronic-circuit", None
106       AdvancedCircuit = "advanced-circuit", None
107       ProcessingUnit = "processing-unit", None
108       EngineUnit = "engine-unit", None
109       ElectricEngineUnit = "electric-engine-unit", None
110       Lab = "lab", Lab
111       Accumulator = "accumulator", Accumulator
112       GunTurret = "gun-turret", GunTurret
113       PiercingRoundsMagazine = "piercing-rounds-magazine", Ammo
114       FirearmMagazine = "firearm-magazine", Ammo
115       Grenade = "grenade", None
116       Radar = "radar", Entity
117       StoneWall = "stone-wall", Entity
118       Gate = "gate", Entity
119       SmallLamp = "small-lamp", Entity
120       NuclearReactor = "nuclear-reactor", Reactor
121       UraniumFuelCell = "uranium-fuel-cell", None
122       HeatExchanger = \'heat-exchanger\', HeatExchanger
123       AutomationSciencePack = "automation-science-pack", None
124       MilitarySciencePack = "military-science-pack", None
125       LogisticsSciencePack = "logistic-science-pack", None
126       ProductionSciencePack = "production-science-pack", None
127       UtilitySciencePack = "utility-science-pack", None
128       ChemicalSciencePack = "chemical-science-pack", None
129
130       ProductivityModule = "productivity-module", None
131       ProductivityModule2 = "productivity-module-2", None
132       ProductivityModule3 = "productivity-module-3", None
133       FlyingRobotFrame = "flying-robot-frame", None
134       RocketSilo = "rocket-silo", RocketSilo
135       Rocket = "rocket", Rocket
136       Satellite = "satellite", None
137       RocketPart = "rocket-part", None
138       RocketControlUnit = "rocket-control-unit", None
139       LowDensityStructure = "low-density-structure", None
140       RocketFuel = "rocket-fuel", None
141       SpaceSciencePack = "space-science-pack", None
142       BeltGroup = "belt-group", BeltGroup
143       PipeGroup = "pipe-group", PipeGroup
144       ElectricityGroup = "electricity-group", ElectricityGroup
145       def __init__(self, prototype_name, entity_class_name):
146           self.prototype_name = prototype_name
147           self.entity_class = entity_class_name
148       @property
```

```python
        def WIDTH(self):
            return self.entity_class._width  # Access the class attribute directly

        @property
        def HEIGHT(self):
            return self.entity_class._height
prototype_by_name = {prototype.value[0]: prototype for prototype in Prototype}
prototype_by_title = {str(prototype): prototype for prototype in Prototype}
class Technology(enum.Enum):
    Automation = "automation"  # Unlocks assembling machine 1
    Automation2 = "automation-2"  # Unlocks assembling machine 2
    Automation3 = "automation-3"  # Unlocks assembling machine 3
    Logistics = "logistics"  # Unlocks basic belts and inserters
    Logistics2 = "logistics-2"  # Unlocks fast belts and inserters
    Logistics3 = "logistics-3"  # Unlocks express belts and inserters
    AdvancedElectronics = "advanced-electronics"
    AdvancedElectronics2 = "advanced-electronics-2"
    Electronics = "electronics"
    ElectricEnergy = "electric-energy-distribution-1"
    ElectricEnergy2 = "electric-energy-distribution-2"
    SolarEnergy = "solar-energy"
    ElectricEngineering = "electric-engine"
    BatteryTechnology = "battery"
    NuclearPower = "nuclear-power"
    SteelProcessing = "steel-processing"
    AdvancedMaterialProcessing = "advanced-material-processing"
    AdvancedMaterialProcessing2 = "advanced-material-processing-2"
    MilitaryScience = "military"
    ModularArmor = "modular-armor"
    PowerArmor = "power-armor"
    PowerArmor2 = "power-armor-mk2"
    NightVision = "night-vision-equipment"
    EnergyShield = "energy-shields"
    EnergyShield2 = "energy-shields-mk2-equipment"
    RailwayTransportation = "railway"
    OilProcessing = "oil-processing"
    AdvancedOilProcessing = "advanced-oil-processing"
    SulfurProcessing = "sulfur-processing"
    Plastics = "plastics"
    Lubricant = "lubricant"
    ProductivityModule = "productivity-module"
    ProductivityModule2 = "productivity-module-2"
    ProductivityModule3 = "productivity-module-3"
    Robotics = "robotics"
    LogisticsSciencePack = "logistic-science-pack"
    MilitarySciencePack = "military-science-pack"
    ChemicalSciencePack = "chemical-science-pack"
    ProductionSciencePack = "production-science-pack"
    FastInserter = "fast-inserter"
    StackInserter = "stack-inserter"
    StackInserterCapacity1 = "stack-inserter-capacity-bonus-1"
    StackInserterCapacity2 = "stack-inserter-capacity-bonus-2"
    StorageTanks = "fluid-handling"
    BarrelFilling = "barrel-filling"
    Grenades = "grenades"
    Landfill = "landfill"
    CharacterInventorySlots = "character-inventory-slots"
    ResearchSpeed = "research-speed"
    SpaceScience = "space-science-pack"
    RocketFuel = "rocket-fuel"
    RocketControl = "rocket-control-unit"
    LowDensityStructure = "low-density-structure"
    RocketSiloTechnology = "rocket-silo"
technology_by_name = {tech.value: tech for tech in Technology}
class Resource:
    Coal = "coal", ResourcePatch
    IronOre = "iron-ore", ResourcePatch
    CopperOre = "copper-ore", ResourcePatch
    Stone = "stone", ResourcePatch
    Water = "water", ResourcePatch
    CrudeOil = "crude-oil", ResourcePatch
    UraniumOre = "uranium-ore", ResourcePatch
    Wood = "wood", ResourcePatch
class EntityStatus(Enum):
    WORKING = \'working\'
    NORMAL = \'normal\'
    NO_POWER = \'no_power\'
    LOW_POWER = \'low_power\'
    NO_FUEL = \'no_fuel\'
    EMPTY = \'empty\'
    NOT_PLUGGED_IN_ELECTRIC_NETWORK = \'not_plugged_in_electric_network\'
```

```python
230        CHARGING = \'charging\'
231        DISCHARGING = \'discharging\'
232        FULLY_CHARGED = \'fully_charged\'
233        NO_RECIPE = \'no_recipe\'
234        NO_INGREDIENTS = \'no_ingredients\'
235        NOT_CONNECTED = \'not_connected\'
236        NO_INPUT_FLUID = \'no_input_fluid\'
237        NO_RESEARCH_IN_PROGRESS = \'no_research_in_progress\'
238        NO_MINABLE_RESOURCES = \'no_minable_resources\'
239        LOW_INPUT_FLUID = \'low_input_fluid\'
240        FLUID_INGREDIENT_SHORTAGE = \'fluid_ingredient_shortage\'
241        FULL_OUTPUT = \'full_output\'
242        FULL_BURNT_RESULT_OUTPUT = \'full_burnt_result_output\'
243        ITEM_INGREDIENT_SHORTAGE = \'item_ingredient_shortage\'
244        MISSING_REQUIRED_FLUID = \'missing_required_fluid\'
245        MISSING_SCIENCE_PACKS = \'missing_science_packs\'
246        WAITING_FOR_SOURCE_ITEMS = \'waiting_for_source_items\'
247        WAITING_FOR_SPACE_IN_DESTINATION = \'waiting_for_space_in_destination\'
248        PREPARING_ROCKET_FOR_LAUNCH = \'preparing_rocket_for_launch\'
249        WAITING_TO_LAUNCH_ROCKET = \'waiting_to_launch_rocket\'
250        LAUNCHING_ROCKET = \'launching_rocket\'
251        NO_AMMO = \'no_ammo\'
252        LOW_TEMPERATURE = \'low_temperature\'
253        NOT_CONNECTED_TO_RAIL = \'not_connected_to_rail\'
254        def __repr__(self):
255        def from_string(cls, status_string):
256        def from_int(cls, status_int):
257   class Inventory(BaseModel):
258        class Config:
259            populate_by_name = True
260            arbitrary_types_allowed = True
261        def __init__(self):
262        def __getitem__(self, key: \'Prototype\', default) -> int:
263        def get(self, key: \'Prototype\', default) -> int:
264        def __setitem__(self, key: \'Prototype\', value: int) -> None:
265        def items(self):
266        def __repr__(self) -> str:
267        def __str__(self) -> str:
268        def __len__(self) -> int:
269        def keys(self):
270        def values(self):
271   class Direction(Enum):
272        UP = 0
273        NORTH = 0
274        RIGHT = 2
275        EAST = 2
276        DOWN = 4
277        SOUTH = 4
278        LEFT = 6
279        WEST = 6
280        def __repr__(self):
281        def from_string(cls, direction_string):
282   class Position(BaseModel):
283        x: float
284        y: float
285        def _parse_positional_args(cls, v):
286        def __init__(self):
287        def parse_args(cls, values):
288        def __hash__(self):
289        def __add__(self, other) -> \'Position\':
290        def __sub__(self, other) -> \'Position\':
291        def is_close(self, a: \'Position\', tolerance: float) -> bool:
292        def distance(self, a: \'Position\') -> float:
293        def _modifier(self, args):
294        def above(self) -> \'Position\':
295        def up(self) -> \'Position\':
296        def below(self) -> \'Position\':
297        def down(self) -> \'Position\':
298        def left(self) -> \'Position\':
299        def right(self) -> \'Position\':
300        def to_bounding_box(self, other: \'Position\') -> \'BoundingBox\':
301        def __eq__(self, other) -> bool:
302   class IndexedPosition(Position):
303        type: str
304        def __new__(cls):
305        def __init__(self):
306        def __hash__(self):
307   class EntityInfo(BaseModel):
308        name: str
309        direction: int
310        position: Position
```

```python
        start_position: Optional[Position]
        end_position: Optional[Position]
        quantity: Optional[int]
        warning: Optional[str]
        contents: Dict[str, int]
        status: EntityStatus
class InspectionResults(BaseModel):
        entities: List[EntityInfo]
        player_position: Tuple[float, float]
        radius: float
        time_elapsed: float
        def get_entity(self, prototype: \'Prototype\') -> Optional[EntityInfo]:
        def get_entities(self, prototype: \'Prototype\') -> List[EntityInfo]:
class BoundingBox(BaseModel):
        left_top: Position
        right_bottom: Position
        left_bottom: Position
        right_top: Position
        def center(self) -> Position:
        def width(self) -> float:
        def height(self) -> float:
class BuildingBox(BaseModel):
        height: int
        width: int
class ResourcePatch(BaseModel):
        name: str
        size: int
        bounding_box: BoundingBox
class Dimensions(BaseModel):
        width: float
        height: float
class TileDimensions(BaseModel):
        tile_width: float
        tile_height: float
class Ingredient(BaseModel):
        name: str
        count: Optional[int]
        type: Optional[Literal[\'fluid\', \'item\']]
class Product(Ingredient):
        probability: Optional[float]
class Recipe(BaseModel):
        name: Optional[str]
        ingredients: Optional[List[Ingredient]]
        products: Optional[List[Product]]
        energy: Optional[float]
        category: Optional[str]
        enabled: bool
class BurnerType(BaseModel):
    """
Type of entity that burns fuel
    """
        class Config:
            arbitrary_types_allowed = True
        fuel: Inventory
class EntityCore(BaseModel):
        name: str
        direction: Direction
        position: Position
        def __repr__(self):
class Entity(EntityCore):
    """
Base class for all entities in the game.
    """
        id: Optional[int]
        energy: float
        type: Optional[str]
        dimensions: Dimensions
        tile_dimensions: TileDimensions
        prototype: Any
        health: float
        warnings: List[str]
        status: EntityStatus
        def __repr__(self) -> str:
        def _get_prototype(self):
        def width(cls):
        def height(cls):
class StaticEntity(Entity):
    """
A static (non-moving) entity in the game.
    """
        neighbours: Optional[Union[Dict, List[EntityCore]]]
```

```
class Rail(Entity):
    """
Railway track for trains.
    """
    _height: float
    _width: float
class Splitter(Entity):
    """
A belt splitter that divides item flow between outputs.
    """
    input_positions: List[Position]
    output_positions: List[Position]
    inventory: List[Inventory]
    _height: float
    _width: float
class TransportBelt(Entity):
    """
A conveyor belt for moving items.
    """
    input_position: Position
    output_position: Position
    inventory: Inventory
    is_terminus: bool
    is_source: bool
    _height: float
    _width: float
    def __repr__(self):
    def __hash__(self):
    def __eq__(self, other):
class Electric(BaseModel):
    """
Base class for entities that interact with the power grid.
    """
    electrical_id: Optional[int]
class ElectricalProducer(Electric, Entity):
    """
An entity that generates electrical power.
    """
    production: Optional[Any]
    energy_source: Optional[Any]
    electric_output_flow_limit: Optional[float]
class EnergySource(BaseModel):
    buffer_capacity: str
    input_flow_limit: str
    output_flow_limit: str
    drain: str
class Accumulator(StaticEntity, Electric):
    """
Represents an energy storage device
    """
    energy_source: Optional[EnergySource]
    _height: float
    _width: float
class Inserter(StaticEntity, Electric):
    """
Represents an inserter that moves items between entities.
    Requires electricity to power
    """
    pickup_position: Optional[Position]
    drop_position: Position
    _width: float
    _height: float
class Filtered(BaseModel):
    filter: Optional[Any]
class UndergroundBelt(TransportBelt):
    """
An underground section of transport belt.
    """
    is_input: bool
    connected_to: Optional[int]
    _height: float
    _width: float
class MiningDrill(StaticEntity):
    """
Base class for mining drills that extract resources.
    The direction of the drill is where the drop_position is oriented towards
    """
    drop_position: Position
    resources: List[Ingredient]
class ElectricMiningDrill(MiningDrill, Electric):
    """
```

```python
An electrically-powered mining drill.
"""
    _height: float
    _width: float
class BurnerInserter(Inserter, BurnerType):
    """
An inserter powered by burnable fuel.
    """
    _height: float
    _width: float
class BurnerMiningDrill(MiningDrill, BurnerType):
    """
A mining drill powered by burnable fuel.
    """
    _width = 2
    _height = 2
class Ammo(BaseModel):
    name: str
    magazine_size: Optional[int]
    reload_time: Optional[float]
class GunTurret(StaticEntity):
    turret_ammo: Inventory
    _height: float
    _width: float
    kills: Optional[int]
class AssemblingMachine(StaticEntity, Electric):
    """
A machine that crafts items from ingredients.
    Requires power to operate
    """
    recipe: Optional[Recipe]
    assembling_machine_input: Inventory
    assembling_machine_output: Inventory
    assembling_machine_modules: Inventory
    _height: float
    _width: float
class FluidHandler(StaticEntity):
    """
Base class for entities that handle fluids
    """
    connection_points: List[Position]
    fluid_box: Optional[Union[dict, list]]
    fluid_systems: Optional[Union[dict, list]]
class AdvancedAssemblingMachine(FluidHandler, AssemblingMachine):
    """
A second and third tier assembling machine that can handle fluids.
    Requires power to operate
    A recipe first needs to be set and then the input fluid source can be connected with pipes
    """
    _height: float
    _width: float
class MultiFluidHandler(StaticEntity):
    """
Base class for entities that handle multiple fluid types.
    """
    input_fluids: List[str]
    output_fluids: List[str]
    input_connection_points: List[IndexedPosition]
    output_connection_points: List[IndexedPosition]
    fluid_box: Optional[Union[dict, list]]
    fluid_systems: Optional[Union[dict, list]]
class FilterInserter(Inserter, Filtered):
    """
A inserter that only moves specific items
    """
    _height: float
    _width: float
class ChemicalPlant(MultiFluidHandler, AssemblingMachine):
    """
Represents a chemical plant that processes fluid recipes.
    Requires powering and accepts input fluids (from storage tanks etc) and solids (with inserters)
    Outputs either:
        solids (battery, plastic) that need to be extracted with inserters
        fluids (sulfuric acid, oil) that need to be extracted with pipes
    IMPORTANT: First a recipe needs to be set and then the fluid sources can be connected to the plant
    """
    _height: float
    _width: float
class OilRefinery(MultiFluidHandler, AssemblingMachine):
    """
An oil refinery for processing crude oil into products.
```

```python
    Requires powering and accepts input fluids (from pumpjacks, storage tanks etc) and solids
    First a recipe needs to be set and then the fluid sources can be connected to the refinery
    """
    _height: float
    _width: float
class PumpJack(MiningDrill, FluidHandler, Electric):
    """
A pump jack for extracting crude oil. Requires electricity
    This needs to be placed on crude oil and oil needs to be extracted with pipes
    Oil can be sent to a storage tank, oil refinery or a chemical plant
    Oil can also be sent to assmbling machine to be made into oil barrels
    Important: The PumpJack needs to be placed on exact crude oil tiles

    """
    _height: float
    _width: float
class SolarPanel(ElectricalProducer):
    """
A solar panel for generating power from sunlight.
    This entity generated power during the day
    Thus it can be directly connected to a entity to power it
    """
    _height: float
    _width: float
class Boiler(FluidHandler, BurnerType):
    """
A boiler that heats water into steam.
    """
    steam_output_point: Optional[Position]
    _height: float
    _width: float
class HeatExchanger(Boiler):
    """
A nuclear heat exchanger that converts water to steam.
    """
class Generator(FluidHandler, StaticEntity):
    """
A steam generator that produces electricity.
    """
    _height: float
    _width: float
class Pump(FluidHandler, Electric):
    """
An electrically-powered fluid pump.
    """
    _height: float
    _width: float
class OffshorePump(FluidHandler):
    """
A pump that extracts water from water tiles.
    Can be used in power generation setups and to supply water to chemical plants and oil refineries.
    """
    _height: float
    _width: float
class ElectricityPole(Entity, Electric):
    """
A power pole for electricity distribution.
    """
    flow_rate: float
    _height: float
    _width: float
    def __hash__(self):
class Furnace(Entity, BurnerType):
    """
A furnace for smelting items
    """
    furnace_source: Inventory
    furnace_result: Inventory
    _height: float
    _width: float
class ElectricFurnace(Entity, Electric):
    """
An electrically-powered furnace.
    """
    furnace_source: Inventory
    furnace_result: Inventory
    _height: float
    _width: float
class Chest(Entity):
    """
A storage chest.
```

```python
    """
        inventory: Inventory
        _height: float
        _width: float
class StorageTank(FluidHandler):
    """
    A tank for storing fluids.
        Can be used for inputs and outputs of chemical plants and refineries.
        Also can store water from offshore pumps.
    """
        _height: float
        _width: float
class RocketSilo(StaticEntity, Electric):
    """
    A rocket silo that can build and launch rockets.
    """
        rocket_parts: int
        rocket_inventory: Inventory
        rocket_progress: float
        launch_count: int
        _width: float
        _height: float
        def __repr__(self) -> str:
class Rocket(Entity):
    """
    A rocket that can be launched from a silo.
    """
        payload: Optional[Inventory]
        launch_progress: float
        def __repr__(self) -> str:
class Lab(Entity, Electric):
    """
    A research laboratory.
    """
        lab_input: Inventory
        lab_modules: Inventory
        research: Optional[Any]
        _height: float
        _width: float
        def __repr__(self) -> str:
class Pipe(Entity):
    """
    A pipe for fluid transport
    """
        fluidbox_id: int
        flow_rate: float
        contents: float
        fluid: Optional[str]
        _height: float
        _width: float
class Reactor(StaticEntity):
    """
    A nuclear reactor
    """
        _height: float
        _width: float
class EntityGroup(BaseModel):
        id: int
        status: EntityStatus
        position: Position
        name: str
class WallGroup(EntityGroup):
    """
    A wall
    """
        name: str
        entities: List[Entity]
class BeltGroup(EntityGroup):
    """
    A connected group of transport belts.
    """
        belts: List[TransportBelt]
        inputs: List[Entity]
        outputs: List[Entity]
        inventory: Inventory
        name: str
        def __repr__(self) -> str:
        def __str__(self):
class PipeGroup(EntityGroup):
    """
    A connected group of pipes.
```

```
716  """
717      pipes: List[Pipe]
718      name: str
719      def __repr__(self) -> str:
720      def __str__(self):
721  class ElectricityGroup(EntityGroup):
722  """
723  Represents a connected power network.
724  """
725      name: str
726      poles: List[ElectricityPole]
727      def __repr__(self) -> str:
728      def __hash__(self):
729      def __str__(self):
730  ‘‘‘
731  ‘‘‘methods
732  can_place_entity(entity: Prototype, direction: Direction = <Direction.UP: 0>, position: Position =
        Position(x=0.0, y=0.0)) -> bool
733  """
734  Tests to see if an entity can be placed at a given position
735  :param entity: Entity to place from inventory
736  :param direction: Cardinal direction to place entity
737  :param position: Position to place entity
738  :return: True if entity can be placed at position, else False
739  """
740
741  craft_item(entity: Prototype, quantity: int = 1) -> int
742  """
743  Craft an item from a Prototype if the ingredients exist in your inventory.
744  :param entity: Entity to craft
745  :param quantity: Quantity to craft
746  :return: Number of items crafted
747  """
748
749  extract_item(entity: Prototype, source: Union[Position, Entity], quantity=5) -> int
750  """
751  Extract an item from an entity\'s inventory at position (x, y) if it exists on the world.
752  :param entity: Entity prototype to extract, e.g Prototype.IronPlate
753  :param source: Entity or position to extract from
754  :param quantity: Quantity to extract
755  :example extract_item(Prototype.IronPlate, stone_furnace.position, 5)
756  :example extract_item(Prototype.CopperWire, stone_furnace, 5)
757  :return The number of items extracted.
758  """
759
760  get_connection_amount(source: Union[Position, Entity, EntityGroup], target: Union[Position, Entity,
        EntityGroup], connection_type: Prototype = <Prototype.Pipe: (\'pipe\', <class \'Pipe\'>)>) -> int
761  """
762  Calculate the number of connecting entities needed to connect two entities, positions or groups.
763  :param source: First entity or position
764  :param target: Second entity or position
765  :param connection_type: a Pipe, TransportBelt or ElectricPole
766  :return: A integer representing how many entities are required to connect the source and target entities
767  """
768
769  get_entities(entities: Union[Set[Prototype], Prototype] = set(), position: Position = None, radius: float
        = 1000) -> List[Entity]
770  """
771  Get entities within a radius of a given position.
772  :param entities: Set of entity prototypes to filter by. If empty, all entities are returned.
773  :param position: Position to search around. Can be a Position object or "player" for player\'s position.
774  :param radius: Radius to search within.
775  :return: Found entities
776  """
777
778  get_entity(entity: Prototype, position: Position) -> Entity
779  """
780  Retrieve a given entity object at position (x, y) if it exists on the world.
781  :param entity: Entity prototype to get, e.g Prototype.StoneFurnace
782  :param position: Position where to look
783  :return: Entity object
784  """
785
786  get_prototype_recipe(prototype: Union[Prototype, RecipeName, str]) -> Recipe
787  """
788  Get the recipe (cost to make) of the given entity prototype.
789  :param prototype: Prototype to get recipe from
790  :return: Recipe of the given prototype
791  """
792
793  get_research_progress(technology: Optional[Technology] = None) -> List[Ingredient]
```

```
794    """
795    Get the progress of research for a specific technology or the current research.
796    :param technology: Optional technology to check. If None, checks current research.
797    :return The remaining ingredients to complete the research
798    """
799
800    get_resource_patch(resource: Resource, position: Position, radius: int = 10) -> Optional[ResourcePatch]
801    """
802    Get the resource patch at position (x, y) if it exists in the radius.
803    if radius is set to 0, it will only check the exact position for this resource patch.
804    :param resource: Resource to get, e.g Resource.Coal
805    :param position: Position to get resource patch
806    :param radius: Radius to search for resource patch
807    :example coal_patch_at_origin = get_resource_patch(Resource.Coal, Position(x=0, y=0))
808    :return: ResourcePatch if found, else None
809    """
810
811    harvest_resource(position: Position, quantity=1, radius=10) -> int
812    """
813    Harvest a resource at position (x, y) if it exists on the world.
814    :param position: Position to harvest resource
815    :param quantity: Quantity to harvest
816    :example harvest_resource(nearest(Resource.Coal), 5)
817    :example harvest_resource(nearest(Resource.Stone), 5)
818    :return: The quantity of the resource harvested
819    """
820
821    insert_item(entity: Prototype, target: Union[Entity, EntityGroup], quantity=5) -> Entity
822    """
823    Insert an item into a target entity\'s inventory
824    :param entity: Type to insert from inventory
825    :param target: Entity to insert into
826    :param quantity: Quantity to insert
827    :return: The target entity inserted into
828    """
829
830    inspect_inventory(entity=None) -> Inventory
831    """
832    Inspects the inventory of the given entity. If no entity is given, inspect your own inventory.
833    :param entity: Entity to inspect
834    :return: Inventory of the given entity
835    """
836
837    launch_rocket(silo: Union[Position, RocketSilo]) -> RocketSilo
838    """
839    Launch a rocket.
840    :param silo: Rocket silo
841    :return: Your final position
842    """
843
844    move_to(position: Position, laying: Prototype = None, leading: Prototype = None) -> Position
845    """
846    Move to a position.
847    :param position: Position to move to.
848    :return: Your final position
849    """
850
851    nearest(type: Union[Prototype, Resource]) -> Position
852    """
853    Find the nearest entity or resource to your position.
854    :param type: Entity or resource type to find
855    :return: Position of nearest entity or resource
856    """
857
858    nearest_buildable(entity: Prototype, building_box: BuildingBox, center_position: Position, **kwargs) ->
               BoundingBox
859    """
860    Find the nearest buildable area for an entity.
861
862    :param entity: Prototype of the entity to build.
863    :param building_box: The building box denoting the area of location that must be placeable.
864    :param center_position: The position to find the nearest area where building box fits
865    :return: BoundingBox of the nearest buildable area or None if no such area exists.
866    """
867
868    pickup_entity(entity: Union[Entity, Prototype, EntityGroup], position: Optional[Position] = None) -> bool
869    """
870    Pick up an entity if it exists on the world at a given position.
871    :param entity: Entity prototype to pickup, e.g Prototype.IronPlate
872    :param position: Position to pickup entity
873    :return: True if the entity was picked up successfully, False otherwise.
```

```
874    """
875
876    place_entity(entity: Prototype, direction: Direction = <Direction.UP: 0>, position: Position =
           Position(x=0.0, y=0.0), exact: bool = True) -> Entity
877    """
878    Places an entity e at local position (x, y) if you have it in inventory.
879    :param entity: Entity to place
880    :param direction: Cardinal direction to place
881    :param position: Position to place entity
882    :param exact: If True, place entity at exact position, else place entity at nearest possible position
883    :return: Entity object
884    """
885
886    place_entity_next_to(entity: Prototype, reference_position: Position = Position(x=0.0, y=0.0), direction:
           Direction = <Direction.RIGHT: 2>, spacing: int = 0) -> Entity
887    """
888    Places an entity next to an existing entity, with an optional space in-between (0 space means adjacent).
889    In order to place something with a gap, you must increase the spacing parameter.
890    :param entity: Entity to place
891    :param reference_position: Position of existing entity or position to place entity next to
892    :param direction: Direction to place entity from reference_position
893    :param spacing: Space between entity and reference_position
894    :example: place_entity_next_to(Prototype.WoodenChest, Position(x=0, y=0), direction=Direction.UP,
           spacing=1)
895    :return: Entity placed
896    """
897
898    print(*args)
899    """
900    Adds a string to stdout
901    :param args:
902    :return:
903    """
904
905    rotate_entity(entity: Entity, direction: Direction = <Direction.UP: 0>) -> Entity
906    """
907    Rotate an entity to a specified direction
908    :param entity: Entity to rotate
909    :param direction: Direction to rotate
910    :example rotate_entity(iron_chest, Direction.UP)
911    :return: Returns the rotated entity
912    """
913
914    set_entity_recipe(entity: Entity, prototype: Union[Prototype, RecipeName]) -> Entity
915    """
916    Sets the recipe of an given entity.
917    :param entity: Entity to set recipe
918    :param prototype: The prototype to create, or a recipe name for more complex processes
919    :return: Entity that had its recipe set
920    """
921
922    set_research(*args, **kwargs)
923    """
924    Call self as a function.
925    """
926
927    set_research(technology: Technology) -> List[Ingredient]
928    """
929    Set the current research technology for the player\'s force.
930    :param technology: Technology to research
931    :return: Required ingredients to research the technology.
932    """
933
934    shift_entity(entity: Entity, direction: Union[Direction, Direction], distance: int = 1) -> Entity
935    """
936    Calculate the number of connecting entities needed to connect two entities, positions or groups.
937    :param source: First entity or position
938    :param target: Second entity or position
939    :param connection_type: a Pipe, TransportBelt or ElectricPole
940    :return: A integer representing how many entities are required to connect the source and target entities
941    """
942
943    sleep(seconds: int) -> bool
944    """
945    Sleep for up to 15 seconds before continuing. Useful for waiting for actions to complete.
946    :param seconds: Number of seconds to sleep.
947    :return: True if sleep was successful.
948    """
949    ```
```

## Example Memory

Below is an example of 2 step memory that is input to the agent. The past policies $P$ and $O$ are formatted like user-assistant messages for chat models, where policies are assistant and observation is user messages. To improve the informativeness of logs, we mask out the overall game setup prints of past policies except for the last step. This makes sure the agent doesnt rely on past game states to make decisions

```
Assstant message
```python
water_pos = nearest(Resource.Water)
print(f"Found water source at {water_pos}")

# Place offshore pump
move_to(water_pos)
offshore_pump = place_entity(Prototype.OffshorePump, position=water_pos)
print(f"Placed offshore pump at {offshore_pump.position}")

boiler = place_entity_next_to(Prototype.Boiler, reference_position=offshore_pump.position, spacing = 4,
    direction = Direction.RIGHT)
print(f"Placed boiler at {boiler.position}")
steam_engine = place_entity_next_to(Prototype.SteamEngine, reference_position=boiler.position, spacing =
    5, direction = Direction.RIGHT)
print(f"Placed steam engine at {steam_engine.position}")
# Connect offshore pump to boiler with pipes
pump_to_boiler = connect_entities(offshore_pump.position, boiler.position, Prototype.Pipe)
print(f"Connected offshore pump to boiler with pipes: {pump_to_boiler}")

# Connect boiler to steam engine with pipes
boiler_to_engine = connect_entities(boiler.position, steam_engine.position, Prototype.Pipe)
print(f"Connected boiler to steam engine with pipes: {boiler_to_engine}")
print(f"Current inventory {inspect_inventory()}")
print(f"Updated entities on the map: {get_entities()}")
```

User message
27015: ('Found water source at x=-12.0 y=0.0',)
27015: ('Placed offshore pump at x=-10.5 y=-0.5',)
27015: ('Placed boiler at x=-5.0 y=-0.5',)
27015: ('Placed steam engine at x=3.5 y=-0.5',)
27015: ('Connected offshore pump to boiler with pipes: [
    PipeGroup(fluid_system=1, position=x=-6.5 y=-2.5, status=EntityStatus.WORKING, pipes=[7 pipes],
    fluid=water)]',)
27015: ('Connected boiler to steam engine with pipes: [ PipeGroup(fluid_system=3, position=x=-2.5 y=-0.5,
    status=EntityStatus.EMPTY, pipes=[5 pipes])]',)
27015: ("Current inventory {'wooden-chest': 10, 'transport-belt': 200, 'burner-inserter': 10,
    'small-electric-pole': 200, 'pipe': 88, 'boiler': 3, 'steam-engine': 1, 'burner-mining-drill': 10,
    'electric-mining-drill': 10, 'offshore-pump': 2, 'stone-furnace': 5, 'assembling-machine-1': 5,
    'coal': 200, 'stone': 50, 'iron-ore': 62, 'iron-plate': 19, 'copper-plate': 52, 'iron-gear-wheel':
    22, 'electronic-circuit': 99}",)
27015: ("Updated entities on the map: <STALE_ENTITY_DATA_OMITTED>")

Assstant message
```python
# move to iron position
iron_pos = nearest(Resource.IronOre)
move_to(iron_pos)
# place a lectric mining drill
drill = place_entity(Prototype.ElectricMiningDrill, position = iron_pos)
print(f"Put a drill to mine iron at {drill.position}")

# power the drill
steam_engine = get_entity(Prototype.SteamEngine, position = Position(x = 3.5, y=-0.5))
poles = connect_entities(steam_engine,drill,Prototype.SmallElectricPole)
print(f"Used poles {poles} to power drill at {drill.position}")

print(f"Current inventory {inspect_inventory()}")
print(f"Updated entities on the map: {get_entities()}")
```

User Message
27015: ('Put a drill to mine iron at x=-11.5 y=19.5',)
27015: ('Used poles [ElectricityGroup(id=18,
    poles=[(x=-1.5,y=4.5),(x=-3.5,y=6.5),(x=3.5,y=1.5),...,(x=-9.5,y=12.5),(x=-11.5,y=14.5)],
    voltage=0)] to power drill at x=-11.5 y=19.5',)
27015: ("Current inventory {'wooden-chest': 10, 'transport-belt': 200, 'burner-inserter': 10,
    'small-electric-pole': 191, 'pipe': 88, 'boiler': 3, 'steam-engine': 1, 'burner-mining-drill': 10,
    'electric-mining-drill': 9, 'offshore-pump': 2, 'stone-furnace': 5, 'assembling-machine-1': 5,
```

```
            'coal': 200, 'stone': 50, 'iron-ore': 62, 'iron-plate': 19, 'copper-plate': 52, 'iron-gear-wheel':
            22, 'electronic-circuit': 99}",)
59  27015: ("Updated entities on the map: [
60  OffshorePump(name='offshore-pump', position=Position(x=-10.5, y=-0.5), direction=Direction.RIGHT,
            energy=0.0, tile_dimensions=TileDimensions(tile_width=1.0, tile_height=1.0),
            status=EntityStatus.WORKING, connection_points=[Position(x=-9.5, y=-0.5)], fluid_box=[{'name':
            'water', 'amount': 100, 'temperature': 15}], fluid_systems=[49]),
61  Boiler(fuel={}, name='boiler', position=Position(x=-5.0, y=-0.5), direction=Direction.RIGHT, energy=0.0,
            tile_dimensions=TileDimensions(tile_width=3.0, tile_height=2.0), warnings=['out of fuel'],
            status=EntityStatus.NO_FUEL, connection_points=[Position(x=-5.5, y=-2.5), Position(x=-5.5, y=1.5)],
            fluid_box=[{'name': 'water', 'amount': 200, 'temperature': 15}], fluid_systems=[49],
            steam_output_point=Position(x=-3.0, y=-0.5)),
62  Generator(electrical_id=18, name='steam-engine', position=Position(x=3.5, y=-0.5),
            direction=Direction.RIGHT, energy=0.0, tile_dimensions=TileDimensions(tile_width=3.0,
            tile_height=5.0), warnings=['not receiving electricity', 'no input liquid', 'No fluid present in
            connections'], status=EntityStatus.NOT_CONNECTED, connection_points=[Position(x=6.0, y=-0.5),
            Position(x=1.0, y=-0.5)], fluid_box=[], fluid_systems=[]),
63  ElectricMiningDrill(electrical_id=18, name='electric-mining-drill', position=Position(x=-11.5, y=19.5),
            direction=Direction.UP, energy=0.0, tile_dimensions=TileDimensions(tile_width=3.0, tile_height=3.0),
            warnings=['not receiving electricity'], status=EntityStatus.NO_POWER,
            drop_position=Position(x=-11.5, y=17.5)),
64  PipeGroup(fluid_system=49, position=x=-6.5 y=-2.5, status=EntityStatus.FULL_OUTPUT, pipes=[7 pipes],
            fluid=water),
65  PipeGroup(fluid_system=51, position=x=-2.5 y=-0.5, status=EntityStatus.EMPTY, pipes=[5 pipes]),
66  ElectricityGroup(id=18,
            poles=[(x=-1.5,y=4.5),(x=-3.5,y=6.5),(x=3.5,y=1.5),...,(x=-9.5,y=12.5),(x=-11.5,y=14.5)],
            voltage=0)]",)
```

