# OpenReview forum: "Factorio Learning Environment"
_NeurIPS.cc/2025/Datasets_and_Benchmarks_Track — NeurIPS 2025 Datasets and Benchmarks Track poster_

### Official Review · Reviewer_K9ia · 2025-07-03

**Rating:** 5
**Confidence:** 3

**Summary:**

This paper presents the Factorio Learning Environment (FLE), a novel open-ended benchmark for evaluating large language models (LLMs) on long-term planning, spatial reasoning, program synthesis, and resource optimization. Built on the game Factorio, FLE introduces two evaluation settings: (1) open-play, where agents aim to build the largest possible factory on a procedurally generated map, and (2) lab-play, consisting of 24 structured, bounded tasks with increasing complexity. The environment uses a Python API for interaction and provides symbolic observations. The paper evaluates six frontier LLMs, including Claude 3.5-Sonnet and GPT-4o, highlighting the limitations of current models in long-horizon planning, spatial reasoning, and error correction. FLE is open-sourced to facilitate further research.

**Additional Feedback:**

* What does the x-axis represent in the first chart in Figure 5?

**Dataset Code Accessibility:**

Yes

**Dataset Code Comments:**

The authors provide a structured codebase with detailed instructions to utilize the code.

**Ethical Considerations:**

No, there are no or only very minor ethics concerns

**Final Justification:**

I appreciate the authors' responses with regards to my concerns on the step limits and the multi-modal capability. I would maintain my score.

**Limitations Weaknesses:**

* In open-play, each agent is limited to 5000 steps, which may be insufficient for evaluating models that exhibit slow but deliberate progress. Notably, some reward trajectories in Figure 3 continue to rise steadily as the 5000-step limit approaches.
*  While spatial reasoning is a core challenge, the paper shows limited benefit from visual input; more systematic exploration of multimodal capabilities could strengthen the conclusions.

**Strengths Contributions:**

* FLE introduces a unique and open-ended agentic benchmark that requires precise factory-scale automation and optimization. The exponentially scaling reward system and unbounded task space ensure that the benchmark remains challenging as models become more capable.
* The combination of open-play and lab-play provides both exploratory and diagnostic perspectives to assess agent behavior and failure modes.
* The authors implement comprehensive experiments and provide both quantitative results and insightful qualitative analysis (e.g., error-correction loops, coding styles, spatial reasoning).
* They provide a clean codebase with detailed instructions.

---

> ### Author Rebuttal · Authors · 2025-07-30
>
> We thank the reviewer for their positive feedback and for recognizing the strengths of our benchmark! Please find your concerns addressed below
>
> ## Fixed number of steps in open-play
> We appreciate the reviewer’s observation. The 5000-step limit corresponds to multiple in-game hours and was chosen to balance computational cost with sufficient temporal horizon for meaningful agent behavior to emerge. Our primary goal was not to evaluate agents under unconstrained time, but to assess competency within a bounded timeframe—a setting more relevant for practical evaluation.
>
> As noted, several models do show steady improvement near the cutoff, which highlights the sensitivity of the benchmark to gradual planning and execution. Importantly, we observed that even within this fixed horizon, the environment was able to distinguish agent capabilities and did not lead to early convergence or saturation in performance. We agree that future work may explore extending the horizon or adapting it per agent strategy, but the current limit effectively supports the core goals of the benchmark, given reasonable resource constraints.
>
> We will make this more clear in the camera-ready version.
>
> ## Figure 5 x-axis
> The x axis in the top part of figure 5 shows the number of distinct antecedent ingredients required to craft an  entity. The higher the number (to the right) the more complex the item. The figure shows how different agents created (or failed to create) more complex entities throughout open-play. We will add this label to the camera-ready version of the paper.
>
> ## Additional multi-modal capability experiments
> We agree that a more systematic exploration of multimodal capabilities would strengthen our conclusions. To this end, we have refined the suite of visual reasoning tasks, and will run additional experiments in visual spatial reasoning for the camera ready version of the paper. Namely:
> ### Basic Entity Recognition Tasks
> **Entity Name Prediction**: Given coordinates, identify what entity exists at that position
>
> **Position Prediction**: Given an entity type, locate its position(s) in the blueprint
>
> **Entity Counting**: Count occurrences of specific entity types
>
> These establish baseline visual recognition capabilities and test whether models can map between visual representations and entity identities/locations.
> ### Spatial Reasoning Tasks
> **Distance Calculations**: Compute Manhattan/Euclidean distances between entities
>
> **Directional Relationships**: Identify relative positions (north/south/east/west)
>
> **Pattern Detection**: Recognize spatial arrangements (lines, grids, clusters)
>
> **Proximity Analysis**: Find nearest/farthest entities or those within a radius
>
> These tasks specifically probe spatial understanding beyond simple recognition, requiring models to reason about geometric relationships in the visual space.
> ### Denoising Tasks
> **Entity Removal**: Remove entities and ask what's missing at specific positions
>
> **Spatial Context Denoising**: Use surrounding entity patterns to identify missing components
>
> These tasks evaluate how well models can visually infer minor flaws in an otherwise working factory.

---

### Official Review · Reviewer_WbQU · 2025-07-03

**Rating:** 5
**Confidence:** 3

**Summary:**

This paper introduces the Factorio learning environment (FLE) for evaluating LLMs inside the game Factorio. FLE introduces both the open-ended task of building the largest factory possible in-game and lab play, where the LLM is put inside a resource-constrained environment with 24 distinct tasks to solve. FLE is studied on several LLMs (both open and closed), including GPT-4o, Claude-3.5-Sonnet, and deepseek-v3. Results demonstrate that current LLMs still struggle with complex automation, planning, and spatial reasoning required by the game, highlighting areas for future LLM development.

**Dataset Code Accessibility:**

Yes

**Dataset Code Comments:**

Repository is well documented and easy to navigate.

**Ethical Considerations:**

No, there are no or only very minor ethics concerns

**Final Justification:**

The rebuttal has addressed my main concerns on the human baseline and the comparison against industrial mods in Minecraft. Therefore I decided to update my score to an accept.

**Limitations Weaknesses:**

- The game uses a very basic step-by-step prompting approach, which could significantly influence the poor performance of the models. For example, past observations of the game that are further away are limited to 1024 tokens. I imagine it is almost impossible to hold all the information of a longer playthrough of Factorio. This leads to the question of how much performance would improve with more sophisticated agent architectures, such as those with more advanced memory systems.
- The Python API interface creates an additional layer of abstraction that does not reflect how humans interact with complex systems. The program-based API forces a much more constrained interaction between the agents and the game, which might make spatial reasoning and fast in-game prototyping harder.
- It is unclear how good the human baseline for the lab-play experiments is, since newer players and experienced players may spend very different amounts of time on each task.
- Table 2 shows that Factorio offers more crafting depth than Minecraft. Does that stay the same if we consider modded Minecraft, such as industrial craft, etc.? If modded Minecraft also provides such depth, what sets the games apart, given that Minecraft is generally more supported by other frameworks/communities?

**Strengths Contributions:**

- The production score objective provides a continuous and unbounded reward signal that can distinguish agents of different capabilities.
- FLE provides a much more sophisticated resource management environment that games like Minecraft cannot offer.
- The paper is well written and easy to follow.

---

> ### Author Rebuttal · Authors · 2025-07-31
>
> Thank you for the thoughtful and constructive review. Below we address each concern with a clear **Claim → Evidence → Take-away** structure and note concrete edits we will make in the camera-ready version.
>
> ---
>
> ## 1) Context length & step-by-step prompting
>
> **Claim.** We intentionally used minimal agent scaffolding in our experiments. We believe this isolates and allows us to evaluate inherent model capabilities rather than prompt engineering and agent scaffolding choices.
>
> **Evidence.**
>
> * All models interact through the same step-wise REPL with identical context limits, making comparisons about the **model** rather than the wrapper.
> * Lab-Play tasks are short and scoped (≤ 128 steps) and do not require more complex memory management to succeed. In our error analysis (Section 4.1), the majority of failures arise from planning, spatial reasoning, and action sequencing—not from missing past context.
> * The agent can store symbolic state in the Python namespace (e.g., entities, coordinates, inventories), define helper functions, and cache intermediate results, mitigating pressure on the context window and providing life-long memory. The agent is aware of the summarisation procedure and can create their own memory structures
>
> **Take-away.** The baseline agent is deliberately simple to avoid confounds; the environment is designed to **invite** stronger agents (RAG, long-term memory, multi-tool planners) but our primary evaluation method ensures that observed differences in performance across FLE stem the **intrinsic capabilities** of current LLMs.
>
> **Planned clarification.** We will add a short paragraph in §3 describing the control rationale; an explicit note in §4 reporting that observed failure modes are reasoning-centric; and a sentence in §6 highlighting that the API exposes simple hooks to implement alternative memory/planning modules (retrieval, vector stores, or multi-agent controllers).
>
> **Additional notes on supporting agent architectures.** We agree that better agents could achieve higher performance in FLE, and in fact, the environment was intentionally designed to support this kind of innovation. We have made it simple for researchers to plug in their own more advanced memory systems, multi-agent protocols, tool-use, and planning abstractions.
> Since submitting this paper to Neurips, we have added explicit support for Gym-style interfaces to our open-source environment and encourage alternative agent implementations.
>
> ---
>
> ## 2) API does not mimic how humans interact with complex systems; does the Python API constrain spatial reasoning?
>
> **Claim.** A programmatic interface is methodologically sound for evaluating LLMs and does not artificially constrain spatial-reasoning.
>
> **Evidence.**
>
> * Many real-world systems are API/DSL-first (cloud orchestration, robotics stacks, CAD/simulation), where agents must reason over **explicit** coordinates, relations, and preconditions/effects.
> * The FLE API requires agents to construct and operate over a symbolic spatial model: coordinate-based placement, connectivity, and validation. We expose queries like `nearest()` and `get_entities()`, and low-level vector operations to reduce arithmetic brittleness while keeping the reasoning burden intact.
> * Our analysis (Section 4.2) shows poor performance on spatial debugging tasks even **with** all necessary information available via the API, indicating that the interface does not trivialize spatial reasoning.
> * Only **15.5%** of failures are attributable to API-usage errors (Section 4.1, Table 1), suggesting the bottleneck is not interface idiosyncrasy but multi-step reasoning and plan execution.
>
> **Take-away.** The API does artificially increase spatial difficulty; it makes it measurable, reproducible, and introspectable—properties essential for a scientific benchmark.
>
> **Planned clarification.** We will (i) explicitly distinguish “information availability” from “reasoning over that information,” (ii) include a short example of coordinate-based diagnosis (API calls + outcome), and (iii) note in §6 that a GUI track is a promising next step; the current paper establishes the code-first foundation.
>
> ---
>
> ## 3) Human baseline in Lab-Play
>
> **Claim.** Our human baseline demonstrates **feasibility under the same constraints** and highlights a knowing-doing gap in current LLM agents.
>
> **Evidence.**
>
> * Procedure: one author with casual (<30 hour) Factorio experience operated **only via the Python API** (no GUI), with the same observation/actions as the LLMs.
> * Result: the human completed substantially more tasks (**20/24**) than any LLM within the per-task step budget, while the best model solved **7/24** tasks.
> * Knowing-doing gap: Many LLMs demonstrated superior encyclopedic knowledge of Factorio mechanics (optimal ratios, automation steps, resource chains) compared to our human operator, yet failed to utilise this knowledge and execute successful multi-step plans. This mirrors the "knowing-doing gap" observed in prior agentic benchmarks (Paglieri et al., BALROG 2024), where LLMs show latent knowledge but struggle with grounded action planning.
> * Error profile: despite weaker game knowledge, the human succeeded through stable plan construction and correction, while models often failed in action sequencing and spatial debugging despite encyclopaedic game knowledge; only **15.5%** of model failures were API-usage errors.
>
> **Take-away.** Tasks are feasible through the API; the primary limitation for LLMs is **grounded multi-step reasoning** rather than interface constraints.
>
> **Planned clarification.** We will tighten §4.1 to (a) clarify the baseline procedure, (b) report human success count and aggregate time/steps, (c) bring out specific examples of the knowing-doing gap we observed and (d) add a small table contrasting human vs. model error categories.
>
> ---
>
> ## 4) Does Table 2’s “crafting depth” advantage survive against **modded Minecraft**?
>
> **Claim.** Yes. Even with popular automation/industry mods, Factorio is better suited for **reproducible, long-horizon, throughput-oriented** evaluation.
>
> **Evidence.**
>
> 1. **Scientific reproducibility.** Factorio is deterministic with stable versioning; modded Minecraft introduces multi-mod compatibility drift, evolving mechanics, and non-deterministic behaviors, complicating comparability and reproduceability required for reliable benchmarks.
> 2. **Native integration with measurable optimization.** Factorio’s belts/inserters/fluids/power are co-designed with consistent timing and diagnostics, yielding clear optimization targets (throughput, queueing, utilization) and constraints.
> 3. **Headroom.** State-of-the-art human Factorio factories scale to **≈ 1 M science/min**, which corresponds to **≈ 2.4 × 10⁶ raw items s⁻¹** flowing through production chains. By contrast, even the most over-engineered IndustrialCraft² megabases top out at ≈ 2 × 10² – 1 × 10³ items s⁻¹ (≈ 10⁴–10⁵ items min⁻¹) of combined input + output. Anything beyond that chokes on the 20-tick-per-second game loop and IC²’s hard “one operation per tick” cap
> 4. **API alignment.** Factorio’s entity model maps cleanly to programmatic state with well-defined preconditions/effects, facilitating deterministic replays, ablations, and error audits.
>
> **Take-away:** For measuring progress, Factorio provides at least **\~5 orders of magnitude** more item-throughput headroom than systems built in IndustrialCraft, enabling finer discrimination (and avoiding saturation) even as models improve by orders of magnitude.
>
> **Planned clarification.** We will (i) add one sentence in §2 summarizing the reproducibility argument, and (ii) include a single-line throughput comparison in the main text with a pointer to a supplemental derivation comparing different game-platforms for system complexity.
>
> ---
>
> ## 5) What the results show
>
> **Claim.** Our results isolate **reasoning & control** limitations, as opposed to insufficient input context being the main bottleneck.
>
> **Evidence:**
> * Failure taxonomy: planning brittleness, spatial misalignment, and inability to repair action sequences dominate, while API misuse accounts for only **15.5%** of failures. These failure modes were seen in both open- and lab-play, where the agent does not require long-term memory management for success
> * Providing all necessary symbolic information (via API queries) does not by itself close the gap, indicating a reasoning rather than perception bottleneck.
>
> **Take-away.** FLE surfaces specific, reproducible failure modes in planning and spatial control that are directly actionable for future agent designs (tool-augmented memory, hierarchical planners, action-space verification).
>
> ---
>
> ## 6) Closing
>
> We appreciate the reviewer’s careful reading. We agree that richer agent architectures (advanced memory, retrieval, multi-agent protocols) are promising and expect them to lift performance on FLE. Our contribution is a **deterministic, introspectable, saturation-resistant** environment and evaluation that cleanly exposes where today’s models struggle. We will make the clarifications above in the camera-ready version. We would deeply appreciate the if the reviewer would reconsider their score in light of our explanations and improvements. Thank you for helping us strengthen the paper.

---

> > ### Comment · Reviewer_WbQU · 2025-08-06
> >
> > Thank you to the authors for taking the time to respond to my comments and address all questions raised in my review. I appreciate the explanation of the knowledge-doing gap and the comparison of item throughput against IndustrialCraft. Overall I'm happy with the response and will adjust my score accordingly.

---

### Official Review · Reviewer_3712 · 2025-07-03

**Rating:** 5
**Confidence:** 2

**Summary:**

The paper proposes the Factorio Learning Environment, which is an open-ended benchmark built on the game Factorio to test long-term planning, automated resource management, and spatial reasoning of language model agents. In the benchmark, the agents need to interact via Python REPL API, giving commands to place entities, move the avatar, and query production statistics. In the benchmark, the models can receive two reward signals: a continuous Production Score that measures item throughput and a discrete Milestone reward for unlocking items or technologies. Factorio Learning Environment includes Open-Play and Lab-Play, which are two different modes in Factorio. During evaluation on the benchmark, Claude 3.5-Sonnet perform best but still struggle with complex automation chains and spatial layout.

**Dataset Code Accessibility:**

Yes

**Dataset Code Comments:**

All code, client, and evaluation suites are publicly released to facilitate reproducibility and community extension.

**Ethical Comments:**

I do not foresee any ethical issue within this paper.

**Ethical Considerations:**

No, there are no or only very minor ethics concerns

**Limitations Weaknesses:**

Please find the weaknesses below:
1. The Python REPL API is different from natural language. It may not fully capture the capacity of reasoning of LLMs in natural language. Moreover, the Python REPL API may provide logs, error traces, and production statistics, which contains far richer information than typical real-world control interfaces.
2. There is no theoretical analysis of the hardness of all types of tasks in Factorio Learning Environment. Providing some complexity classification or equivalence results may help.
3. For the evaluation part, while a human benchmark is reported for planning, no human F1 score is given for the five spatial reasoning tests. Moreover, it seems that all experiments use one procedurally generated map seed, leaving generalization across diverse layouts unverified.

**Strengths Contributions:**

Please find the strengths below:
1. Factorio Learning Environment provides an unbounded environment that grows in complexity, which improves upon the fixed-state limitations of prior agent benchmarks. Moreover, it has both Production Score and Milestone rewards which balance long-term optimization with stage-wise progress, a novel combination in agentic benchmarks.
2. In Factorio Learning Environment, agents interact via a live Python REPL, giving commands and defining in-session functions or classes as symbolic memory. This design enables rich, programmatic control not found in most static benchmarks.
3. Factorio Learning Environment includes both Open-Play and Lab-Play, covering both unbounded factory building and targeted sub-challenges. This dual setup offers a framework with multiple choices of evaluated aspects.

---

> ### Author Rebuttal · Authors · 2025-07-30
>
> We thank the reviewer for their thoughtful feedback and recognizing the strengths of FLE. Please find our clarifications to the weaknesses below
>
> ## Python REPL API design decisions
>
> This concern raises an important question about interface modality and its influence on the kinds of reasoning we evaluate. We acknowledge that the FLE agent interface is not “pure” natural language—but we argue that this is a feature, not a flaw, for the following reasons:
>
> ### Agents Can Freely Mix Natural Language and Code
> The REPL interface explicitly supports both natural language and Python. LLMs can use # comments, """ docstrings """, and external think-aloud blocks to reason in natural language before generating actions. In practice, our baseline agents do reason in natural language—often writing plans or strategies before executing them or writing comments and docstrings during python generation.
>
> ### The Rich Observations Are Intentional
> While the REPL provides structured logs, production stats, and error traces, this is by design. The goal is to evaluate how well agents can utilize structured observations, much like in real developer environments or tool-augmented agents (e.g., SWEBench, Gorilla, OpenInterpreter), where the agents receive rich outputs (error logs, print statement results)  of their actions and tool calls. These logs require multi-step reasoning to interpret (e.g., inferring bottlenecks or errors in production chains, root cause analysis) and the success depends on interpreting the semantics and dynamics of the environment. This does mimic how real world engineers and operators work as many real-world control interfaces (e.g., production pipelines, dashboards, APIs) expose programmatic or symbolic abstractions (and dense error logs), not plain language.
>
> FLE is not meant to replace pure NL tasks, but to expand the frontier of what agent reasoning benchmarks look like. We believe it moves beyond static QA or text-only games and into structured interaction, tool-use, and planning.
>
> ## Theoretical analysis of lab-play tasks
> We thank the reviewer for bringing out this gap in our paper. Naturally, we can analyse the difficulty of the lab tasks by looking at the raw resource complexity and required crafting steps of the target item. Raw resource complexity measures how many resources are required to craft the specific item and required crafting steps bring out how many intermediate items are needed to create, i.e how many distinct factory areas and machines are required in the final factory. Figure 4 did take this into account and classifies some of the early-game tasks into distinct buckets of difficulty but we will elaborate on this further in paper and bring this analysis out as an appendix.
>
> We will develop this further for the camera ready version, by including cyclometric complexity as a metric for how hard tasks are, i.e how complex a factory needs to be in order to complete a task.
>
> ## Human baseline for Spatial Reasoning tasks
> Thank you for raising your concern of the lack of human F1 accuracy for the 5 spatial reasoning tasks. We have run the experiments and the average human baseline f1 score for the 5 reasoning tasks is 0.764. While this shows the difficulty of the tasks (not acquiring close to 1), there are still clear gaps between the best performing model (Claude 3.5 Sonnet - 0.24) and human performance on our visual domain
>
> ## Fixed map seed for evaluations
> While our experiments were carried out in one map, that was also by design. Our goal was to minimise confounding variables during evaluation and use environments where the focus was more on factory-optimisation and less on navigating difficult environments.
>
> However, FLE natively supports procedurally generated environments so experimenting with various map seeds is simple. We ran our open-play experiments on a real seeded Factorio map, and lab-play in a human-crafted scenario (providing nearby access to all raw resources).
>
> ## Conclusion
> We're grateful for the reviewer for the constructive feedback. We've made substantial improvements to our paper based on your comments and feel these changes have enhanced the quality of our work. We would appreciate the consideration of a higher confidence rating or further guidance on what additional improvements would be necessary.

---

### Official Review · Reviewer_Y4ZG · 2025-07-07

**Rating:** 5
**Confidence:** 3

**Summary:**

This paper uses a very interesting game environment (Factorio, a strategical game to build your own factory) to test LLM agents' abilities in long-term planning, spatial reasoning, program synthesis, and resource optimization. This is fun, and also the experiment are prudently conducted. The paper is also well written. But the main questions are (1) since the game is not built by the authors, will it be counted as a contribution under the definition of Benchmark and Dataset Track? (2) what will be the real-world contribution of this benchmark.

**Dataset Code Accessibility:**

Yes

**Dataset Code Comments:**

the Python API, client, and interactive environment are available in the github

**Ethical Considerations:**

No, there are no or only very minor ethics concerns

**Final Justification:**

After rebuttal, I raised my score

**Limitations Weaknesses:**

This is the weakness:

### Dataset and Benchmark

W1: Since the game, Factorio, is not built by the authors, and instead, the authors only provide the Python API, client, and interactive environment to interact with Factorio: so I am not sure if it can be counted as a contribution under the definition of Benchmark and Dataset Track?

### Model Selections

W2: I believe the recent Reasoning Language Model (such as Qwen-3-think, DeepSeek-R1, and more) may easily solve the "complex" and strategical tasks due to their strong internal reasoning abilities. So I suggest that the authors can categorize the models into two (1) non-reasoning, and (2) reasoning models, and compare the differences of their performances and behaviours in Factorio. This will bring more interesting insights.

### Questions:

W3: In the long-functional calling and planning, what we observed, which is also reported in [R1], is LLM's cognitive burden: with the longer functional calling procedures, the LLMs get lazier for the next round tool calling and result in premature termination of the task. I see similar discussion in this paper. Can you also maybe elaborate more based on the reference [R1]?

[R1] PENCIL: Long Thoughts with Short Memory

### Applicability and Generalizability:

W4: Moreover, the game is fun. But what will be the real-world applicability and contribution of this paper? Since we all know there will be huge sim-to-real gap.

W5: Besides, so far, all the insights are based on the sole game Factorio, will they be general enough?

**Strengths Contributions:**

S1: The setting is fun and intersting. Factorio, such a strategical game, is indeed a nice exam to  test LLM agents' abilities in long-term planning, spatial reasoning, program synthesis, and resource optimization.

S2: The tested LLMs are complete, the authors tested Claude 3.5, GPT-4o, GPT-4o-Mini,  Deepseek-v3, Gemini-2, and Llama-3.3-79B.

S3: The paper is also well-written, with important insights cearly highlighted for the readers. And the insights are also justified by the experiments based on Factorio.

---

> ### Author Rebuttal · Authors · 2025-07-30
>
> We thank the reviewer for their insightful comments and recognizing the important evaluation aspects of FLE. We address your specific concerns below:
> ## W1: Does building on Factorio qualify as a benchmark contribution?
> The NeurIPS 2025 Datasets & Benchmarks (D&B) call explicitly lists “data generators and reinforcement-learning environments” as in-scope.
>
> FLE is a new, carefully designed reinforcement learning environment that contributes:
> - a structured Python API with custom observation/action spaces
> - task suites designed for evaluating LLMs across planning, spatial reasoning, and tool use
> - a REPL agent interface, curated task curriculum, and scoring metrics
> - Is fully reproducible, open-source implementation, that is designed for controlled evaluation
>
> Many accepted benchmark environments in past NeurIPS and ML conferences have been built on top of existing games, physics engines, or simulators, including Minecraft (e.g., MineDojo, MineRL) and Starcraft, for instance in the 2024 D&B Benchmark track the following paper was published built on top of Starcraft II:
>
> - JaxMARL: Multi‑Agent RL Environments and Algorithms in JAX - This paper presents JaxMARL, a GPU-accelerated codebase and benchmark suite that includes environments built on the StarCraft Multi-Agent Challenge (SMAC), which itself is based on StarCraft II .
>
> Therefore we believe our work falls within the scope of 2025 Neurips Datasets and Benchmarks track.
> ## W2 - Reasoning models
> This is a valid concern, which we appreciate you raising. When composing this paper, we intentionally left out reasoning models from our open-play evaluations, due to the high cost of these experiments. However, to address your concern, we have begun running additional reasoning lab-play experiments on o3, R1 and Qwen-3-think:
>
> Preliminary results on reasoning models:
> - o3: 7/24 tasks completed (same as Claude 3.5), produces steel/plastics but not circuits
> - DeepSeek-R1: 6/24 tasks (in progress)
> - Qwen-3-think: Results pending (ETA: 2 days)
>
> Our initial findings suggest reasoning models don't overcome the fundamental bottlenecks we identify. Full results with error analysis will be in camera-ready.
>
> ## W3 - Cognitive burden during long function-calling (cf. R1).
> We appreciate the reviewer’s insightful connection to [R1], which highlights the limitations of long-horizon CoT reasoning due to suboptimal context usage and growing cognitive burden. We indeed observed similar degradation effects in early experiments: as agent interactions progressed, large amounts of stale or irrelevant context (e.g., past environment observations or generated code) would accumulate, leading to premature task abandonment or repetitive errors.
>
> We mitigated these effects through two primary mechanisms:
>
> - Function Abstraction and Reuse
> The Python API supports agent-defined functions, allowing models to encapsulate and reuse common operations and abstractions (e.g., factory area placement, raw resource gathering, environment exploration). This significantly reduces the need for repetitive low-level code generation. Once a utility function is defined, the model can refer back to it rather than re-implementing logic from scratch—analogous to cognitive chunking that reduces planning burden across long-horizon tasks. This approach offers an avenue for agents to continually improve - while maintaining a similar cognitive burden.
>
> - Context Summarization and Selective Masking
> To reduce unnecessary cognitive load, we recursively summarize agent actions, the environment feedback of actions and errors from earlier steps (more than 16 steps into the past) into a report, focusing on agent learning from past exploration of the action space. The report also includes the signature (input/output variables and docstring) of any historical agent defined functions to ensure the agent can re-use past programs, functions and classes.  Moreover, we selectively mask outdated / stale environment feedback from the current context to minimise the agent acting on outdated information. This ensures the model focuses on relevant, up-to-date state information without being distracted by obsolete details, mirroring the context reduction principles introduced in [R1].
>
> Together, these techniques alleviate the memory strain associated with long sequences of code and tool usage, and help the agent maintain coherence and momentum throughout more complex tasks. We agree with the reviewer that further improvements in context management—such as dynamic memory buffers or learned context pruning—represent promising directions for future work in this setting. We will make the rationale for the design decisions above more clear for the camera ready version.
>
> ## W4/W5: What will be the real-world applicability and contribution of this paper?
>
> We appreciate these important questions about generalizability and real-world applicability. Our key insight is that FLE tests fundamental reasoning capabilities that transfer broadly, not game-specific skills.
>
> ### The Capabilities We Measure Are Domain-General
> The bottlenecks we observe through analysing agent interactions within our environment- spatial reasoning, error correction, and long-horizon planning - represent core limitations observed in LLM research that manifest across domains:
>
> - Error Correction: Zhang et al. (2025, R1) show LLMs fail at intrinsic self-correction across reasoning tasks, while Olausson et al. (2024, R2) find limitations in self-error correction of coding models in HumanEval and APPS tasks
> - Long-Horizon Planning: Xie et.al (2024, R3) show that GPT-4o only succeeds in 0.6% multi-step travel planning tasks while Dagan et al. (2025, R4) demonstrate long-horizon planning limitations in multi-step reasoning tasks
> - Spatial Reasoning: Jia et al. (2025, R5) show that existing frontier models still exhibit significant limitations in simple spatial-reasoning tasks
>
> We have designed FLE to not test "Factorio mastery" but provide a controlled testbed where these fundamental capabilities can be systematically evaluated and their interactions studied.
>
> R1 - Understanding the Dark Side of LLMs’ Intrinsic Self-Correction
>
> R2 - Is Self-Repair a Silver Bullet for Code Generation?
>
> R3 - TravelPlanner: A Benchmark for Real-World Planning with Language Agents
>
> R4 - Plancraft: an evaluation dataset for planning with LLM agents
>
> R5 - OmniSpatial: Towards Comprehensive Spatial Reasoning Benchmark for Vision Language Models
>
> ### Simulation as Scientific Method
> We agree that the sim-to-real gap exists but believe it doesn't invalidate simulation-based evaluation, but is an invaluable tool in agentic research and is a standard scientific approach. Major breakthroughs (AlphaFold, OpenAI Five, autonomous systems) emerged from simulated environments that enabled controlled experimentation. FLE's value lies in:
> - Scalable complexity: Unlike static benchmarks, agents face exponentially scaling realistic challenges that reveal capability limits
> - Systematic evaluation: We can isolate and measure specific reasoning failures in a reproducible environment
> - Diagnostic power: The environment reveals why agents fail, not just that they fail
>
> ### Broader Impact
> Rather than claiming direct real-world transfer, we position FLE as advancing our scientific understanding of LLM limitations in reasoning in complex systems. The insights about spatial planning, error recovery, and system design inform development of more capable agents across domains.
>
> Future work should indeed validate these findings across multiple environments - we see FLE as one component of a broader evaluation ecosystem for agentic capabilities.
>
> ## Conclusion
> Thank you for your thoughtful and valuable feedback. We hope that if we have alleviated your concerns, you might be kind enough to favourably reconsider your score. We're grateful for your constructive feedback to our paper.

---

> > ### Author Response · Authors · 2025-08-07
> >
> > Dear reviewer Y4ZG,
> >
> > We hope we have addressed all your concerns in our response. We would appreciate if you would let us know if you have any more questions, concerns or aspects that need clarification. Thank you!

---

### Decision · Program_Chairs · 2025-09-18

**Decision:**

Accept (poster)

**Comment:**

Following in a long line of research that turns games into benchmarks, this does so for Factorio (an economics simulator where one must balance a dynamical system's inputs, intermediate steps, and outputs).

We've seen significant advances in ML from games. And this is a new challenge compared to what we already have available in the community. Compared to Minecraft this provides orders of magnitude more challenge in terms of what can be crafted.

The manuscript shows that when models attempt to play the game they encounter significant difficulties. One downside of the manuscript is that it connects to the game through an API, an interface that humans don't use.

Given the popularity of environments based on games like Minecraft and how these environments have pushed the community to develop new algorithms and approaches this is likely to be a useful contribution.